# Proton-transfer-induced 3D/2D hybrid perovskites suppress ion migration and reduce luminance overshoot

Hobeom Kim [1,17], Joo Sung Kim [1,17], Jung-Min Heo [1,17], Mingyuan Pei[2], In-Hyeok Park [3], Zhun Liu[4], Hyung Joong Yun[5], Min-Ho Park[1], Su-Hun Jeong[1], Young-Hoon Kim[1], Jin-Woo Park [1], Emad Oveisi [6], Satyawan Nagane[7], Aditya Sadhanala[7,8,16], Lijun Zhang [4], Jin Jung Kweon[9], Sung Keun Lee [9,10], Hoichang Yang[2], Hyun Myung Jang [11], Richard H. Friend [7], Kian Ping Loh [3], Mohammad Khaja Nazeeruddin [12], Nam-Gyu Park [13] & Tae-Woo Lee [1,14,15 ✉]

Perovskite light-emitting diodes (PeLEDs) based on three-dimensional (3D) polycrystalline perovskites suffer from ion migration, which causes overshoot of luminance over time during operation and reduces its operational lifetime. Here, we demonstrate 3D/2D hybrid PeLEDs with extremely reduced luminance overshoot and 21 times longer operational lifetime than 3D PeLEDs. The luminance overshoot ratio of 3D/2D hybrid PeLED is only 7.4% which is greatly lower than that of 3D PeLED (150.4%). The 3D/2D hybrid perovskite is obtained by adding a small amount of neutral benzylamine to methylammonium lead bromide, which induces a proton transfer from methylammonium to benzylamine and enables crystallization of 2D perovskite without destroying the 3D phase. Benzylammonium in the perovskite lattice suppresses formation of deep-trap states and ion migration, thereby enhances both operating stability and luminous efficiency based on its retardation effect in reorientation.

[1] Department of Materials Science and Engineering, Seoul National University, Seoul 08826, Republic of Korea. [2] Department of Chemical Engineering, Inha University, Incheon 22212, Republic of Korea. [3] Department of Chemistry, National University of Singapore, Singapore 117543, Singapore. [4] State Key Laboratory of Integrated Optoelectronics, Key Laboratory of Automobile Materials of MOE and College of Materials Science and Engineering, Jilin University, Changchun 130012, China. [5] Research Center for Materials Analysis, Korea Basic Science Institute (KBSI), Daejeon 34133, Republic of Korea. [6] Interdisciplinary Centre for Electron Microscopy (CIME), École Polytechnique Fédérale de Lausanne (EPFL), Lausanne CH-1951, Switzerland. [7] Cavendish Laboratory, University of Cambridge, JJ Thomson Avenue, Cambridge CB3 0HE, UK. [8] Clarendon Laboratory, Department of Physics, University of Oxford, Parks Road, Oxford OX1 3PU, UK. [9] School of Earth and Environmental Sciences, Seoul National University, Seoul 08826, Republic of Korea. [10] Institute of Applied Physics, Seoul National University, Seoul 08826, Republic of Korea. [11] Department of Materials Science and Engineering, Pohang University of Science and Technology, Pohang 37673, Republic of Korea. [12] Group for Molecular Engineering of Function Materials, Institute of Chemical Sciences and Engineering, École Polytechnique Fédérale de Lausanne (EPFL), Sion CH-1951, Switzerland. [13] Department of Materials Science and Engineering, Sungkyunkwan University, Suwon 16419, Republic of Korea. [14] School of Chemical and Biological Engineering, Seoul National University, Seoul 08826, Republic of Korea. [15] Institute of Engineering Research, Research Institute of Advanced Materials, Nano Systems Institute (NSI), Seoul National University, Seoul 08826, Republic of Korea. [16]Present address: Centre for Nano Science and Engineering, Indian Institute of Science, Bangalore 560012, India. [17]These authors contributed equally: Hobeom Kim, Joo Sung Kim, Jung-Min Heo. ✉email: twlees@snu.ac.kr

I n the early days of the research on perovskites light-emitting diodes (PeLEDs), three-dimensional (3D) polycrystalline perovskites have been employed as a light-emitter based on the advantages such as high colour purity, facile tuning of emission colour, low material cost and low-temperature processability[1–3]. Extensive research has contributed to rapid increase in luminous efficiency of PeLEDs by various approaches, such as introduction of nanocrystal pinning (NCP) process, management of compositional distribution, incorporation of polymer into perovskite and an enhancement in light outcoupling[3–6]. However, 3D polycrystalline perovskite emitters have a fundamental limitation in efficient radiative recombination because of the high density of trap states which originates primarily from ionic defects and their migration degrades the operating lifetime of PeLEDs; this problem must be overcome before they can be practically used for lighting and displays.

There have been many reports on PeLEDs regarding overshoot of luminance during the initial stage of operation[4,6–8]. However, the mechanism of this phenomenon has not been identified. The ion migration can much negatively influence on device stability because the PeLEDs operate under much higher applied electric field compared with perovskite solar cells. Therefore, to improve the device lifetime of PeLEDs, ion migration must be suppressed[7,9–12]. Especially, the ion migration mostly occurs on grain boundaries (GBs), thus blocking possible migration pathways along the GB can be an effective strategy to suppress it[13].

In this study, we develop a proton-transfer-induced 3D/2D hybrid perovskite emitter that can be an ideal configuration to suppress the ion migration and significantly reduce the overshoot of luminance with improved operational stability in PeLEDs during the initial stage. In a conventional way to prepare low-dimensional layered perovskites, an ammonium halide salt with a bulky backbone (e.g. phenethylammonium halide (PEAX), or $n$-butylammonium halide ($n$-BAX)) and the salts of a 3D perovskite (e.g. methylammonium halide (MAX) and lead halide (PbX$_2$)) have been dissolved together in a solvent or a mixed solvent[8,14–19]. Here, unlike the conventional way, we incorporate a liquid-form neutral reagent, benzylamine (BnA) into the methylammonium lead bromide (MAPbBr$_3$) precursor to form 3D/2D hybrid perovskites[20]. Therefore, the synthetic procedure which requires strong hydrohalic acid reagents to obtain the organic

ammonium halide salt for 2D perovskites becomes unnecessary. To compare with BnA, we use aniline (ANI) as it has the same molecular structure except for a methylene bridge between benzene ring and amine. $^1$H nuclear magnetic resonance (NMR) spectroscopy revealed that the strongly basic BnA participates in the crystallization of 2D perovskite by being transformed into BnA$^+$ by proton-transfer from MA$^+$ in the precursor. Grazing-incident x-ray diffraction (GIXD) technique confirmed the formation of 2D BnA$_2$PbBr$_4$, dominantly with $n = 1$, while retaining the 3D MAPbBr$_3$ phase. In contrast, ANI which has a similar molecular structure with BnA does not work as a reagent because it is weakly basic, so it cannot induce the formation of additional 2D perovskite. The 3D/2D hybrid perovskite film showed a significant improvement in photoluminescence (PL) characteristics compared to the 3D perovskite film because the 2D perovskite can passivate deep trap sites, as well as shallow traps in 3D perovskite, and thereby lead to efficient radiative recombination. Therefore, PeLEDs with the 3D/2D hybrid perovskite emitter had highly improved electroluminescence (EL) efficiency. Furthermore, the 3D/2D hybrid PeLEDs showed a dramatic enhancement in the long-term stability owing to the suppressed ion migration, which led to an extremely reduced initial luminance overshoot ratio as 7.4%. In contrast, the 3D PeLEDs which suffer from ion migration showed a steep increase in initial luminance with the overshoot ratio as 150.4%. The 3D/2D hybrid PeLED had more than 21 times longer operational lifetime ($T_{40} = 810$ min) than 3D PeLEDs ($T_{40} = 38$ min). We cohesively link the PL improvements and the suppression of ion migration to the retardation effect of BnA$^+$ reorientation in the perovskite lattice.

## Results

**Protonation of BnA enabling 2D perovskite formation.** The incorporation of BnA into MAPbBr$_3$ precursor leads to the formation of 3D/2D hybrid dimensional perovskite in which the formation of 2D perovskite does not degrade the existing 3D phase so that 3D and 2D perovskites coexist (Fig. 1). In contrast, the addition of ANI does not make the structural change maintaining the 3D phase of perovskite as the case without an additive. We performed $^1$H NMR spectroscopy to understand the underlying chemistry in MAPbBr$_3$ precursor according to the addition of 2.4 mol% of ANI or BnA compared to MAPbBr$_3$ without the addition before the crystallization of perovskites occurs (Fig. 2a). The precursors were dissolved in dimethyl sulfoxide-$d_6$ (DMSO-$d_6$) as the same condition of the films for actual PeLEDs which we will discuss in the following sections. The spectrum of pristine MAPbBr$_3$ precursor showed two dominant proton signals, one at $\delta = 7.44$ ppm that represents the ammonium (–NH$_3^+$) of MA$^+$, and the other at 2.32 ppm that represents its methyl (–CH$_3$) group. The addition of BnA to the precursor of MAPbBr$_3$ caused an upfield shift of the proton signal of the ammonium group to $\delta = 7.31$ ppm. The proton signal of the ammonium group progressively showed further upfield shift as the amount of BnA addition increased; the signal reached 6.33 ppm at 50 mol% BnA and 5.67 ppm at 100 mol% BnA (Fig. 2b). These changes can be attributed to the strong shielding of the ammonium moiety of MA$^+$ by BnA reagent. In contrast, the addition of ANI to the MAPbBr$_3$ precursor did not cause any shift. Even high concentrations of ANI did not cause a noticeable chemical shift of the proton signal of MA (Fig. 2c); this lack of effect indicates that the solution with ANI remained as a mixture without causing a chemical reaction even though ANI has the same molecular structure as BnA except the methylene bridge (–CH$_2$) between the amine and the benzene ring. The integration ratio ($I$) of proton signals ($H_{ammonium}/H_{alkyl}$) increased from 1 for pristine MAPbBr$_3$ to 1.81 for MAPbBr$_3$ with 100 mol% of BnA

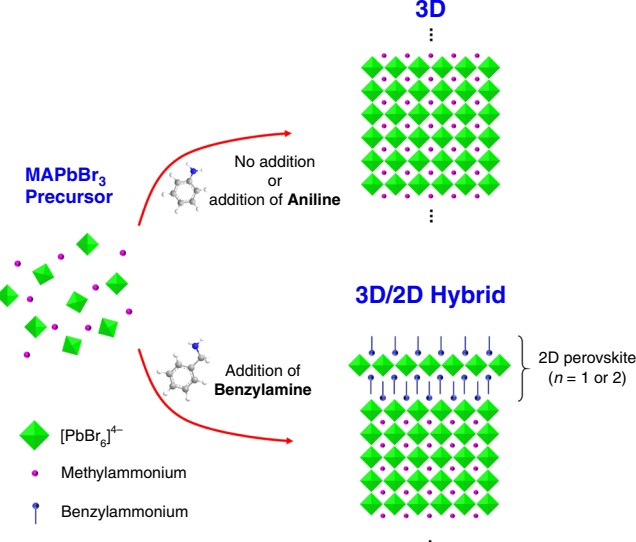

**Fig. 1 Principal scheme of our study.** The neutral reagent, benzylamine (BnA) leads to the formation of 3D/2D hybrid perovskites while aniline (ANI) does not induce the formation of 2D perovskites.

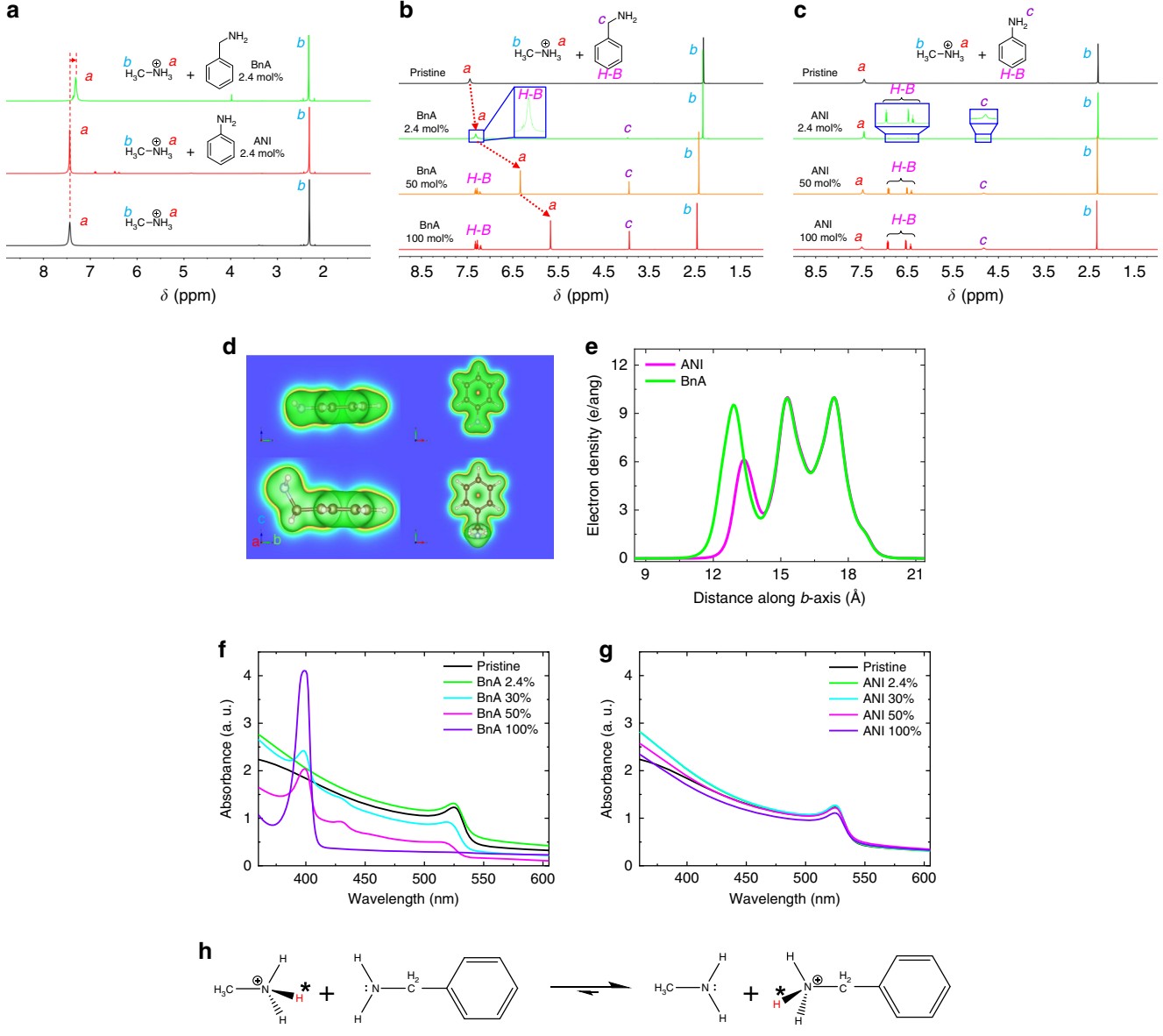

**Fig. 2 Proton-transfer-induced 2D perovskite formation with BnA. a** [1]H NMR spectra of MAPbBr$_3$ precursor solutions in DMSO-$d_6$ according after addition of additives. Pristine MAPbBr$_3$ (black), MAPbBr$_3$ with 2.4 mol% of ANI (black), and MAPbBr$_3$ with 2.4 mol% of BnA (green). **b**, **c** [1]H NMR spectra according to the different amount of BnA and ANI addition. **d** Electron density contour plots of BnA (bottom) and ANI (top) with an iso-surface of 0.03 e Å$^{-3}$. **e** Integrated electron density profile for BnA (green) and ANI (red) along with *b*-axis in **d**. **f**, **g** Absorption spectra of MAPbBr$_3$ thin film according to the addition of BnA and ANI. **h** Proton-transfer reaction between MA and BnA.

(Supplementary Fig. 1 and Supplementary Table 1). This change indicates the presence of a strong interaction that represents an increased hydrogen bond strength by sharing a proton between BnA and MA$^+$. In contrast, MAPbBr$_3$ with 100 mol% ANI had $I$ around 1 implying that ANI does not interact with MA$^+$.

These different effects of BnA and ANI on the chemical behaviour can be explained by the Brønsted–Lowry acid–base reaction. BnA is a strong base, so a proton from the ammonium group of MA$^+$ which is a weak acid, can be transferred to nitrogen of BnA to yield BnA$^+$. In contrast, ANI is a weak base, so it cannot extract a proton from MA$^+$. To learn the cause of this difference, we calculated the electron densities of BnA and ANI with the first-principles density functional theory (DFT). The resultant contour plots of the electron density of each molecule and its integrated profiles along the *b*-axis direction are presented (Fig. 2d, e). The calculated results indicate that BnA has

a much higher electron density on the terminal amine moiety than ANI because the methylene bridge on BnA can donate more electrons to the more electronegative nitrogen and localise the electrons on the nitrogen leading to an inductive effect. ANI, in contrast, has electrons that are delocalized between the nitrogen and the benzene ring governed by a resonance effect, which makes ANI much less basic than BnA. As a consequence of these differences, the protonation tendency from MA$^+$ to BnA is $5.5 \times 10^4$ times stronger than from MA$^+$ to ANI (Supplementary Note 1).

The absorption characteristics of the solid-state perovskite films from the corresponding precursors support this result (Fig. 2f, g): the excitonic peak of the MAPbBr$_3$ absorption spectrum at 525 nm changes significantly according to the amount of BnA added, but does not shift with the addition of ANI regardless of the amount. For example, the addition of 30 mol% BnA resulted in

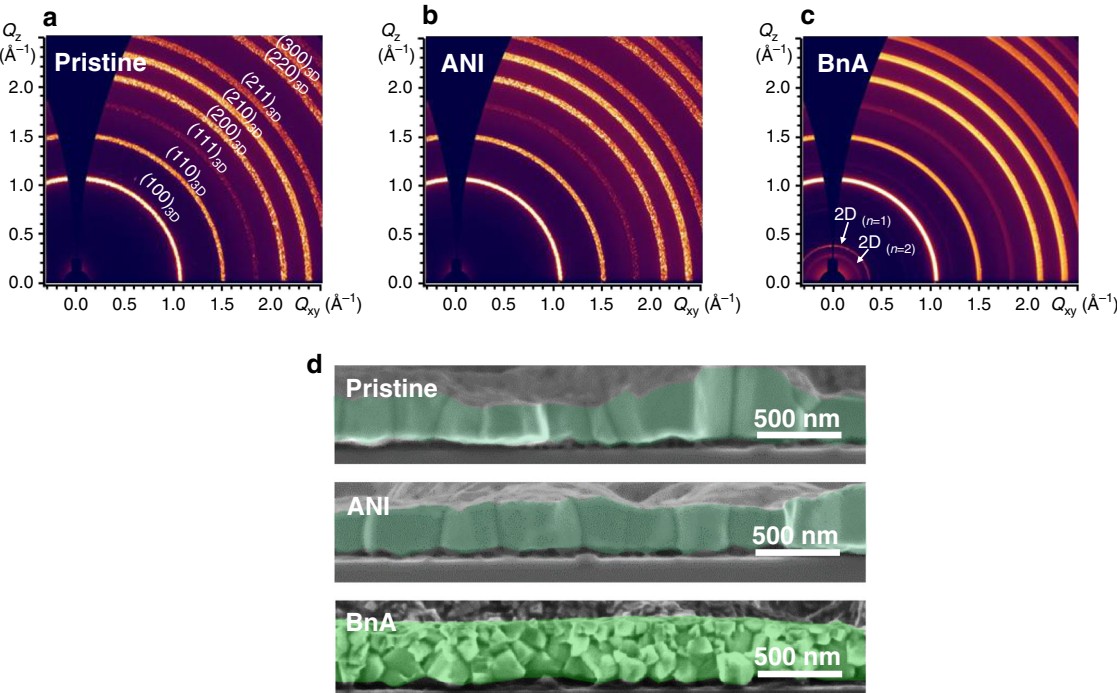

**Fig. 3 Crystal structure and morphology of MAPbBr₃ according to the addition of ANI or BnA. a–c** 2D GIXD patterns of pristine MAPbBr$_3$, MAPbBr$_3$ with 2.4 mol% ANI, and 2.4 mol% BnA. **d** Cross-sectional SEM images of pristine MAPbBr$_3$ (top), MAPbBr$_3$ with 2.4 mol% ANI (middle), and 2.4 mol% BnA (bottom).

new excitonic peaks at 399 and 430 nm, which can be assigned to a phase with $n = 1$ and 2, respectively, indicating the formation of 2D perovskites[21–24]. Interestingly, the film obtained using a precursor with 100 mol% of BnA showed the spectrum of a typical 2D perovskite film that has an excitonic peak at 399 nm; this peak is the result of the strong proton-withdrawing tendency of BnA which takes protons from MA$^+$ and participates in the formation of 2D perovskite (Fig. 2h). Furthermore, we predicted the importance of the protonation of BnA by performing first-principles molecular dynamics (MD) simulations in which perovskite lattice including the protonated BnA (BnA$^+$) showed much greater structural stability than that with neutral BnA. (Supplementary Fig. 2). Also, we performed the solid-state magic-angle-spinning (MAS) NMR spectroscopy to gain an atomic-level insight about the formation of crystalline perovskite (Supplementary Fig. 3). The result confirmed that the protonated BnA composed the solid-state crystalline perovskite while ANI did not participate in the formation of perovskite.

**Crystal structure and configuration of perovskites.** To examine the detail of perovskite crystal structures and dimensionality, we performed grazing-incidence X-ray diffraction (GIXD) on the perovskite films. The results confirmed that BnA contributed to the formation of 2D perovskite but that ANI did not; this result is consistent with the conclusions drawn from the $^1$H NMR analysis and optical absorption spectra. The films of 3D pristine MAPbBr$_3$ and MAPbBr$_3$ with 2.4 mol% ANI showed similar diffraction patterns; this result indicates that the ANI does not influence the perovskite crystal structure. In contrast, the addition of 2.4 mol% BnA led to an appearance of additional peaks at low angles; the most dominant one was at $2\theta = 5.32°$ (Supplementary Fig. 4). The newly appeared peak can be assigned to a 2D perovskite with the lattice plane spacing of 16.58 Å according to Bragg's equation, which corresponds to BnA$_2$PbBr$_4$ with $n = 1$ (Table S2). The information about the crystal structure of perovskite also can be

shown in 2D GIXD patterns (Fig. 3a–c). It is clear that the MAPbBr$_3$ film with 2.4 mol% BnA had an additional X-ray reflection ring at scattering vector $Q = 0.38$ Å$^{-1}$; this is a consequence of the 2D phase. In contrast, the MAPbBr$_3$ with 2.4 mol% ANI had the same pattern as the pristine film of the 3D phase. It is worth noting that MAPbBr$_3$ with BnA retained the diffraction patterns of 3D phase; i.e., 3D MAPbBr$_3$ and 2D BnA$_2$PbBr$_4$ coexisted in the film, which can be called 3D/2D hybrid perovskite. Ring-shaped diffraction patterns with a uniform intensity indicate a random orientation and can be attributed to a NCP process that renders grains small by rapid crystallization of perovskite[25]; such small grains can induce an efficient radiative recombination[3,26–28]. An increase in the ratio of BnA to 30 mol% increased the intensity of the signal of the 2D perovskite phase, whereas the increase in the amount of ANI addition did not cause the formation of any other phase while maintaining 3D phase (Supplementary Fig. 4).

The addition of BnA caused an apparent change in the film morphology of perovskites. The films of pristine MAPbBr$_3$ and MAPbBr$_3$ with 2.4 mol% ANI had a similar columnar grain structure, whereas 3D/2D hybrid perovskite film with 2.4 mol% BnA had granular grains with a smaller size (Fig. 3d). This is because the growth of 2D perovskite can decrease the growth rate of 3D perovskite leading to the formation of a smaller granular type of grains, which can be beneficial to efficient exciton confinement within a grain[3,26]. The morphology of 3D/2D hybrid perovskite films with BnA showed a strong dependence on the amount of BnA (Supplementary Fig. 5). However, the addition of even a large amount of ANI had little influence on the morphology of MAPbBr$_3$ film.

**Improvements in PL characteristics and trap passivation.** We investigated PL characteristics of the 3D and the 3D/2D hybrid perovskite films. The perovskite films were prepared on a glass substrate/buffer hole-injection layer (Buf-HIL), which consists of

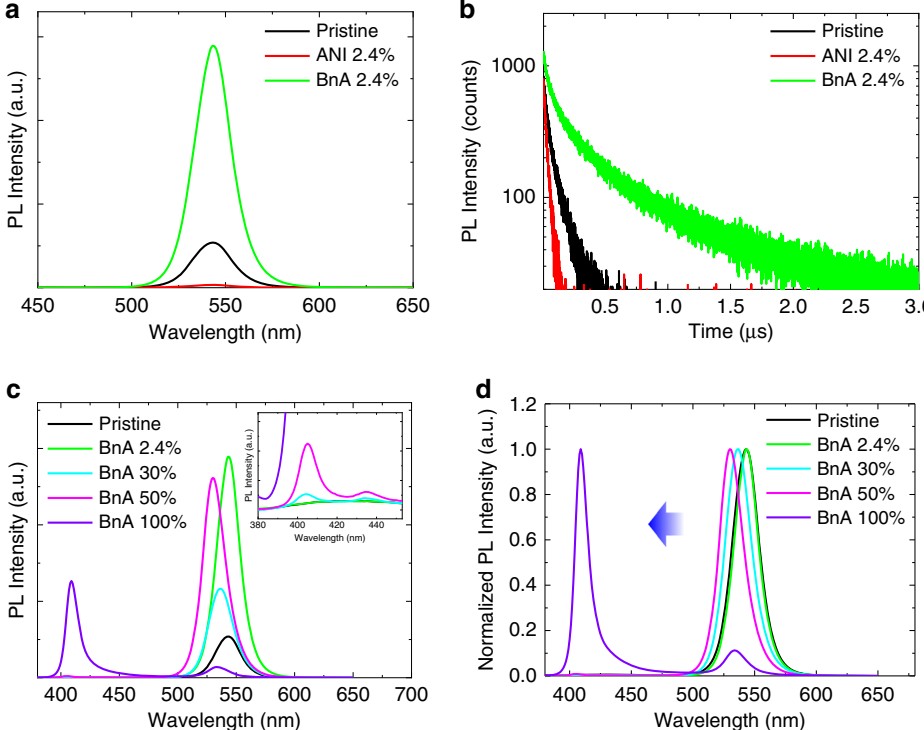

**Fig. 4 PL characteristics of perovskites and the trap passivation. a** PL spectrum of pristine MAPbBr$_3$ film (black), film with 2.4 mol% of ANI (red), and 2.4 mol% of BnA (green). **b** PL lifetime curves of pristine MAPbBr$_3$ film (black), film with 2.4 mol% of ANI (red), and 2.4 mol% of BnA (green). **c** PL spectra of MAPbBr$_3$ film with different amount of BnA and magnified PL spectra at low wavelength region (inset). **d** Normalized PL spectra of MAPbBr$_3$ film with different amount of BnA.

poly(3,4-ethylenedioxythiophene) polystyrene sulfonate (PEDOT: PSS) and perfluorinated ionomer. In steady-state PL measurement, the 3D/2D hybrid perovskite film with 2.4 mol% BnA showed a much higher PL intensity than the 3D perovskite film, whereas the addition of 2.4 mol% ANI caused significant PL quenching (Fig. 4a). Also, the transient PL measurement of the films resulted in the same trend as in the steady-state PL (Fig. 4b). The PL average lifetime $\tau_{Ave}$ was more than four times longer in 3D/2D hybrid perovskite film ($\tau_{Ave} = 254.4$ ns) than in pristine 3D film ($\tau_{Ave} = 60.94$ ns), but the film with ANI showed considerably shortened PL lifetime ($\tau_{Ave} = 24.71$ ns). We summarized the detailed decay parameters based on the bi-exponential decay function in Supplementary Table 3. We realized that 2.4 mol% BnA did not cause a shift in the peak position compared to the PL of 3D MAPbBr$_3$ at 543 nm, or cause the formation of 2D perovskite PL emission peaks. These observations indicate that the radiative recombination occurs only in the 3D perovskite rather than in the 2D perovskite, which can be attributed to the energy cascade that results from the difference in the dimensionality between perovskite phases in the 3D/2D hybrid structure (Fig. 4c, d)[16,29]. Only the addition of the larger amount of BnA caused a blue-shift of PL emission (to 537 nm for 30 mol% and to 530 nm for 50 mol%), accompanied by the appearance of 2D perovskite PL emissions at 405 and 435 nm, which can be assigned to $n = 1$ and 2, respectively. The further addition of BnA up to 100 mol% which corresponds to the same amount as MA, caused a dramatic increase in the intensity of PL emission at 409 nm from 2D perovskite. This shift can be more clearly seen in normalized PL spectra (Fig. 4d). We also measured the absolute PL quantum efficiency (PLQE) which showed a similar trend to the steady-state PL intensity according to the amount of BnA (Supplementary Fig. 6). On the other hand, increasing the

amount of ANI only decreased the PL intensity, and did not shift the emission wavelength, which can be attributed to intensive non-radiative recombination due to the electron deficiency of ANI (Supplementary Fig. 7)[30].

Also, we showed that alignment of energy bands contributes to the improvement in PL properties of the 3D/2D hybrid perovskite film based on ultraviolet photoemission spectroscopy (UPS) analysis (Supplementary Figs. 8, 9 and Supplementary Note 2). We further investigated the carrier confinement effect in the perovskites by performing the excitation density-dependent PL measurement at room temperature (Supplementary Fig. 10). In the excitation density range of $10^{15}$–$10^{18}$ cm$^{-3}$ which corresponds to the operation regime of LEDs[31], the integrate PL (PL$_{int}$) follows a power-law dependence on the excitation density, $n_{ex}$ as PL$_{int} \propto n_{ex}^k$ where the 3D perovskite had $k = 1.30$ indicating the co-existence of excitons (monomolecular emission) and free carriers (bimolecular emission), while the 3D/2D hybrid perovskite film showed $k = 1.08$ indicating the predominant excitonic character by monomolecular recombination, which can be attributed to the effective carrier confinement[32].

To deeply understand the trap passivation effect by using 3D/2D hybrid perovskite, we investigated temperature-dependent steady-state PL characteristics of the perovskite films from 70 to 300 K. The 3D perovskite film showed two distinct PL emission peaks at low temperatures below 130 K (Fig. 5a). While the sharp PL peaks with higher energy can arise from band-edge (BE) emission, the broad PL band with lower energy can be attributed to shallow trap states induced by structural disorder in the orthorhombic phase of MAPbBr$_3$[33,34]. In this temperature regime, the ratio of integrated PL intensity of BE versus the shallow trap-mediated PL emission (TE) was increasing as temperature increased, which implies that thermally assisted

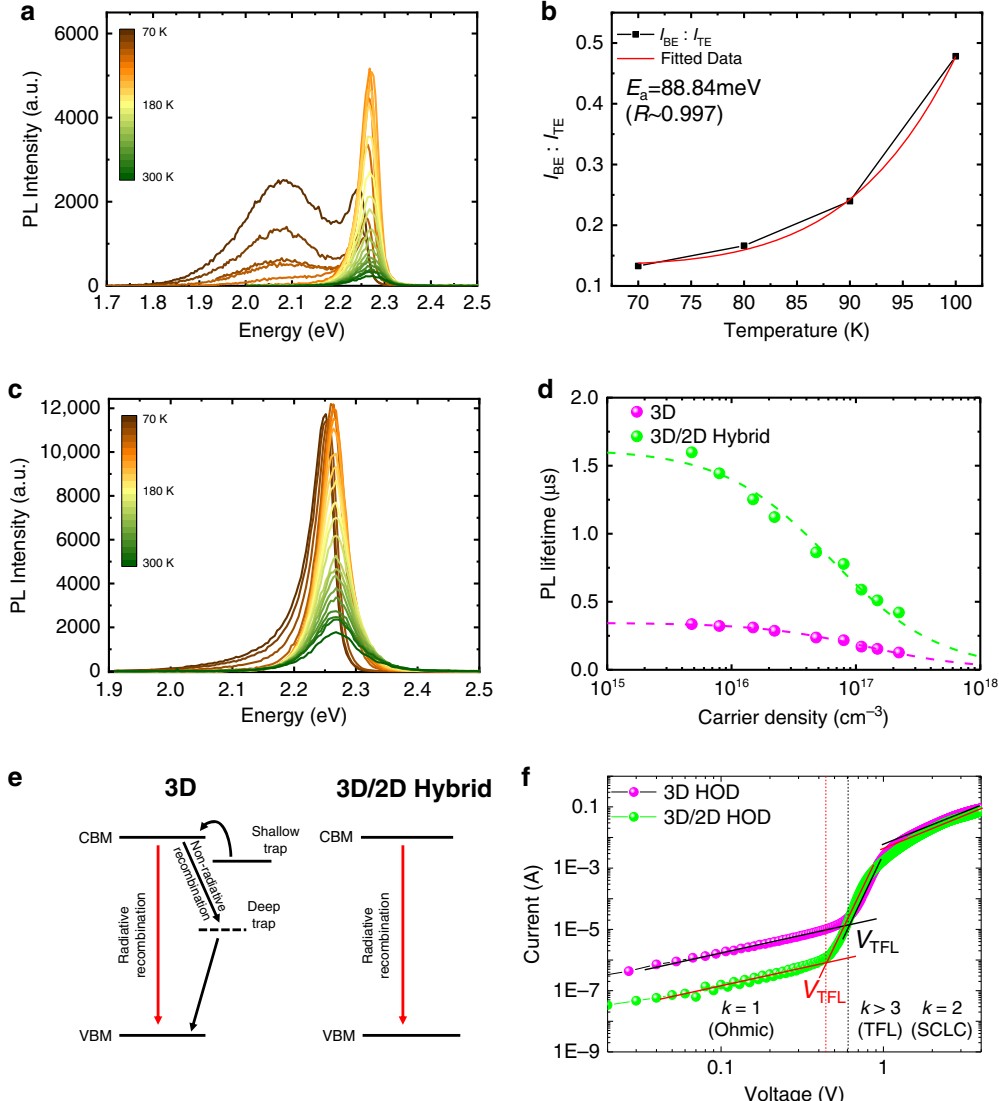

**Fig. 5 Photo-physical analyses to investigate recombination kinetics and trap states of perovskites. a** Temperature-dependent PL of 3D perovskite and **b** Ratio between integrated PL of band emission and trap-mediated emission of 3D perovskite as a function of temperature and its fitted curve according to Boltzmann distribution function. **c** Temperature-dependent PL of 3D/2D hybrid perovskite. **d** Excited charge carrier density-dependent PL lifetimes of the perovskites. **e** Schematic of charge carrier recombination dynamics of the perovskites. **f** I–V curves of hole-only devices with 3D perovskite and 3D/2D hybrid perovskite.

de-trapping of charge carriers occurs (Fig. 5b and Supplementary Fig. 11). To estimate the value of the activation energy to de-trap the charge carriers from the shallow traps, the curve was fitted by the Boltzmann distribution function:

$$y = y_0 e^{\left(-\frac{E_a}{k_B T}\right)} \tag{1}$$

where $k_B$ is the Boltzmann constant and $T$ is temperature and $E_a$ is the activation energy to de-trap charge carriers, which resulted in 88.84 meV. In contrast, the 3D/2D hybrid perovskite film only showed a single PL peak of BE without the TE in the whole range of temperature (Fig. 5c). This indicates that the incorporation of 2D perovskites effectively passivates the shallow trap states even in the orthorhombic phase.

Furthermore, the excitation density dependence of the PL lifetimes revealed defect-related recombination kinetics of the perovskites (Fig. 5d). We controlled the excited charge carrier density of the perovskite films from $4.8 \times 10^{15}$ to $2.2 \times 10^{17}$ cm$^{-3}$ (Supplementary Fig. 12). We used a rate equation model where

the decay of the excited charge carrier density $n_0$ can be explained regarding radiative and non-radiative recombination, which can be written as

$$\tau_{PL} = (A + B n_0)^{-1} \tag{2}$$

where $A$ is the non-radiative recombination rate coefficient, and $B$ is the radiative recombination rate coefficient. We found that the 3D/2D hybrid perovskite film exhibited a significantly lower $A = 6.16 \times 10^5$ s$^{-1}$ than that of 3D perovskite ($A = 2.96 \times 10^6$ s$^{-1}$). This indicates that the 3D perovskite has a high density of non-radiative Shockley–Read–Hall recombination centre which can be associated with certain kinds of ionic defects in the lattice despite a high degree of defect tolerance of perovskite[35]. In contrast, the incorporation of 2D perovskite into 3D perovskite can restrict the defect formation and migration leading to such lowered non-radiative recombination rate coefficient. This is due to the presence of BnA$^+$ which has a lower rotational degree of freedom than MA$^+$ [36], which will be discussed in detail in the following section. Meanwhile, the 3D perovskite and the 3D/2D hybrid

perovskite showed a similar radiative recombination rate coefficient, $B$ as $2.30 \times 10^{-11}$ and $1.08 \times 10^{-11}$ cm$^3$ s$^{-1}$, respectively. This result well matches with the explanation that the shallow traps mostly disappeared in both perovskites. On the basis of the results of the comprehensive PL analyses, we depicted a schematic of the charge recombination dynamics of 3D and 3D/2D hybrid perovskites (Fig. 5e). In the 3D perovskite, the excited charge carriers can be trapped into the shallow traps in the orthorhombic phase, which can be de-trapped by increased thermal energy[33]. Even though the structural stabilization by the phase transition of MAPbBr$_3$ can lead to the reduction in the density of the activated shallow trap states, deep level traps associated with ionic defects can still exist and cause non-radiative recombination. In contrast, 3D/2D hybrid perovskite can have more efficient radiative recombination because the incorporation of 2D perovskite can lead to effective passivation by suppressing the formation and migration of the ionic defects thereby minimizing a loss of charge carrier from quenching by the deep traps.

To verify the reduced density of trap states in 3D/2D hybrid perovskite compared to 3D perovskite, we designed and characterized hole-only devices (HODs) with a structure of ITO/Buf-HIL (50 nm)/Perovskite (400 nm)/MoO$_3$ (10 nm)/Au (100 nm) (Fig. 5f). The I–V behaviour of the devices had three regimes defined by slope $k$: an Ohmic regime ($k = 1$), a trap-filled limited (TFL) regime ($k > 3$) and a trap-free space charge limited current (SCLC) regime ($k = 2$) as the bias increased[37–39]. Here, trap density $n_t$ is linearly proportional to trap-filled limit voltage $V_{TFL}$ at which a transition of I–V behaviour from Ohmic to TFL occurs:

$$V_{TFL} = n_t \frac{eL^2}{2\varepsilon\varepsilon_0} \qquad (3)$$

where $e$ is the electron charge, $L$ is the thickness of the perovskites, $\varepsilon$ is the relative dielectric constant (25.5 for MAPbBr$_3$), and $\varepsilon_0$ is the vacuum permittivity[40]. In the I–V curves, $V_{TFL}$ was lower in HODs based on 3D/2D hybrid perovskite (0.436 V) than in HODs based on pristine 3D perovskite (0.609 V). Thus, we obtained $n_t = 9.49 \times 10^{15}$ cm$^{-3}$ for 3D/2D hybrid perovskite and $n_t = 1.33 \times 10^{16}$ cm$^{-3}$ for 3D perovskite. This result supports that the formation of 2D perovskite can passivate the trap states of 3D perovskite leading to efficient radiative recombination. We also calculated the hole mobility $\mu_h$ of each device by fitting the curves in the regime of trap-free SCLC based on the Mott–Gurney law:

$$J = \frac{9\varepsilon\varepsilon_0\mu_h V^2}{8L^3} \qquad (4)$$

where $V$ is applied voltage. The hole mobility of two devices had similar value $\mu_h$: 0.042 cm$^2$ V$^{-1}$ s$^{-1}$ in the HOD with 3D/2D hybrid perovskite, and 0.049 cm$^2$ V$^{-1}$ s$^{-1}$ in the HOD device with 3D perovskite. This result indicates that the formation of 2D perovskite does not degrade hole transport in the perovskite.

**EL characteristics of PeLEDs and ion migration**. We used the developed perovskite films as an emitter in PeLEDs. The device structure was FTO/Buf-HIL (50 nm)/perovskite (400 nm)/2,2′,2″-(1,3,5-benzinetriyl)-tris(1-phenyl-1-H-benzimidazole) (TPBi) (50 nm)/LiF (1 nm)/Al (100 nm) (Fig. 6a). All LED characteristics were significantly improved in the device with the 3D/2D hybrid perovskite emitter compared to those of the device with the 3D perovskite emitter. The 3D/2D PeLED had maximum current efficiency CE$_{max}$ = 20.55 cd A$^{-1}$, and external quantum efficiency EQE$_{max}$ = 4.23%, which are both higher than in the 3D PeLED (CE$_{max}$ = 13.23 cd A$^{-1}$ and EQE$_{max}$ = 2.50%) (Fig. 6b–e). Both

devices had the same normalized EL emission spectra, which is consistent with the result of PL spectra (Fig. 6f). Devices were optimized by adjusting the amount of BnA (Supplementary Fig. 13 and Supplementary Table 5). We attribute the improvement in EL characteristics of 3D/2D hybrid PeLED to the efficient exciton confinement and the reduced defect-assisted recombination that result from the passivation of trap sites by the formation of 2D perovskite[31]. The effective suppression of ion migration by the 3D/2D hybrid perovskite structure can also contribute to the EL improvement because charge carriers can result in emission of light instead of going through non-radiative recombination by the ionic space charges that are distributed at the vicinity of perovskite interfaces[41]. In contrast, the use of ANI-added perovskite film degraded the LED characteristics due to significant non-radiative recombination by the electron-deficient ANI as well as by the ionic space charges at the interfaces with the perovskite-emitting layer (Supplementary Fig. 14).

To verify that there is a different behaviour regarding ion migration between 3D/2D hybrid and 3D PeLEDs, we observed a time-dependent change in current under constant voltage (CV), and time-dependent change in voltage under constant current (CC). Under CV of 3.5 V, the 3D/2D hybrid PeLED maintained relatively stable current during measurement (Fig. 7a); this result is a consequence of suppressed ion migration, so the local electric field at the interfaces remains mild. In contrast, the 3D PeLED showed severe overshoot of current during the initial stage of the measurement; we attribute this phenomenon to rapidly facilitated charge injection that occurs because the accumulated ions at the interfaces cause narrowing of the depletion region width or lowering of the injection barrier[42].

We also examined a voltage change of PeLEDs by varying the applied CC from 0.16 to 2.22 mA cm$^{-2}$ step-wise by repeating switching on and off (Fig. 7b, c). The 3D/2D hybrid PeLED showed stable plateaus of voltage; this phenomenon proves that the device does not suffer severely from ion migration and accumulation. In contrast, the 3D PeLED showed a voltage undershoot at the beginning of each step, then recovery after the saturation threshold of ion migration. The undershoot of voltage can be explained by the facilitated charge injection, which instantly decreases the built-in potential of the device. The time to reach the saturation threshold of ion migration decreased from 25 to 5 s as CC was increased from 0.16 to 1.11 mA cm$^{-2}$, but the further increase in CC did not cause undershoot in voltage; this result indicates that the migration can be accelerated by increasing the applied current. Magnification (Fig. 7c) of a part of the steps with the CC of 1.11 mA cm$^{-2}$ clearly shows the difference in the stabilities of the devices and peculiar behaviour of the 3D PeLED.

To understand the behaviour of ion migration, we investigated the frequency-dependent capacitance of both types of PeLED in darkness (Fig. 7d). Both devices showed similar capacitance curves in the frequency range 100 Hz $\leq f \leq$ 1 MHz, in which capacitance is a result of geometric parameters and dipolar polarization, but 3D PeLED showed increasing capacitance at $f <$ 100 Hz. This additional capacitance may originate from mobile ions that accumulate at contact interfaces because of the severe ion migration[43,44]. The 3D/2D hybrid PeLED did not show the additional capacitance; this result indicates that ion migration was effectively suppressed.

Here, we propose possible pathways of ion migration in 3D and 3D/2D hybrid perovskite and a mechanism of ion migration suppression in the latter (Fig. 7e). The diffusion barrier energy within the 3D bulk region (Fig. 7e, red) can be considered similar in both systems because they have similar formation energy of ion defects in the 3D bulk. However, the barrier energy for ion diffusion becomes highly different when the migrating ions reach

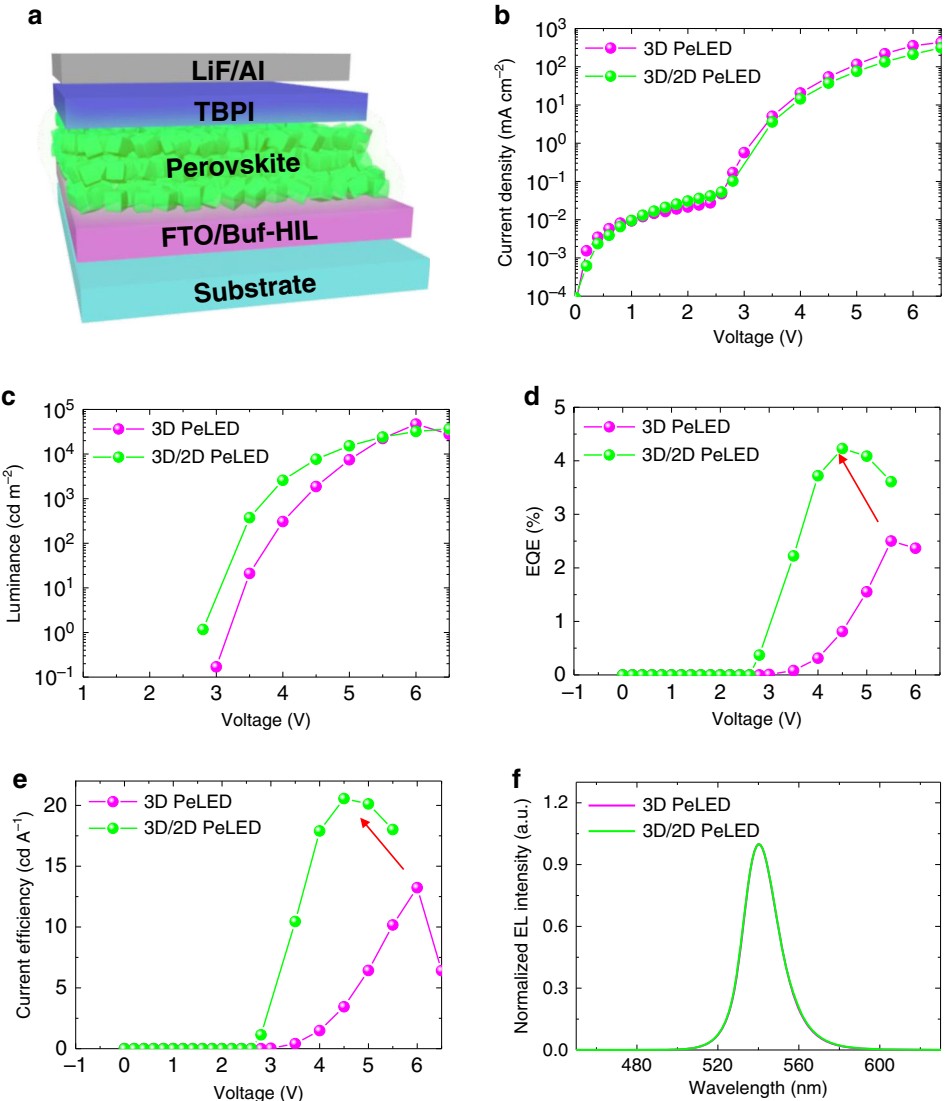

**Fig. 6 EL characteristics of PeLEDs. a** Device structure of PeLEDs. **b** Current density of PeLEDs as a function of applied voltage. **c** Luminance of PeLEDs as a function of applied voltage. **d** External quantum efficiency (EQE) of PeLED as a function of applied voltage. **e** Current efficiency (CE) of PeLEDs as a function of applied voltage. **f** Normalized EL spectra of PeLEDs.

the GB, where ion migration most easily occurs (Fig. 7e, green). In 3D perovskite, the ions can reach the GB without being required to diffuse across the 2D perovskite barrier layer. In 3D/2D hybrid perovskite, on the contrary, the ions should overcome the higher barrier energy to get over the organic capping in the 2D perovskite region (Fig. 7e, purple). We summarized the possible pathways of ion migration in the perovskites and presented a qualitative comparison of the energy barrier for ion migration between 3D perovskite and 3D/2D hybrid perovskite (Supplementary Table 6). Also, the 3D/2D hybrid perovskite can effectively suppress the ion migration due to the morphological advantage observed by scanning electron microscopy (SEM) images (Fig. 3d). As the ion migration mostly occurs on GBs, mobile ions have to migrate much longer pathway in the granular-like 3D/2D hybrid perovskite than in the columnar-like 3D perovskite (Supplementary Fig. 15). Furthermore, the 3D/2D hybrid perovskite has a much greater number of boundary nodes where migrating ions along the direction of the external electric field can be blocked and lose their kinetic energy leading to the termination of their migration. In order to scrutinize the structure

of the 3D/2D hybrid perovskites in grain scale, we used high-angle annular dark-field scanning transmission electron microscopy (HAADF-STEM) and clearly observed the 2D perovskite layer with $n = 1$ (Supplementary Fig. 16).

Ab initio MD simulations predicted that ion migration in perovskites can be greatly assisted by $MA^+$ reorientation in response to motion of the halide ion[36]. For example, the rotation of $MA^+$ can lead to the displacement of a halide ion along the direction of the C–N axis of $MA^+$ based on their attractive Coulomb force. Thus, the incorporation of less orientationally mobile cations with a lower rotational degree of freedom than $MA^+$ can reduce ion migration in perovskites. In this regard, $BnA^+$ can have remarkably reduced reorientation rate by themselves based on their π–π interaction and steric repulsion as a result of neighbouring benzene rings; this slowing of $BnA^+$ reorientation in the perovskite lattice can effectively impede ions migration along GBs compared to $MA^+$ [45].

Combining the simulation results with the transition-state theory of rate processes, we can predict the retardation effect of $BnA^+$ relative to $MA^+$ on the ion migration rate[45]. We

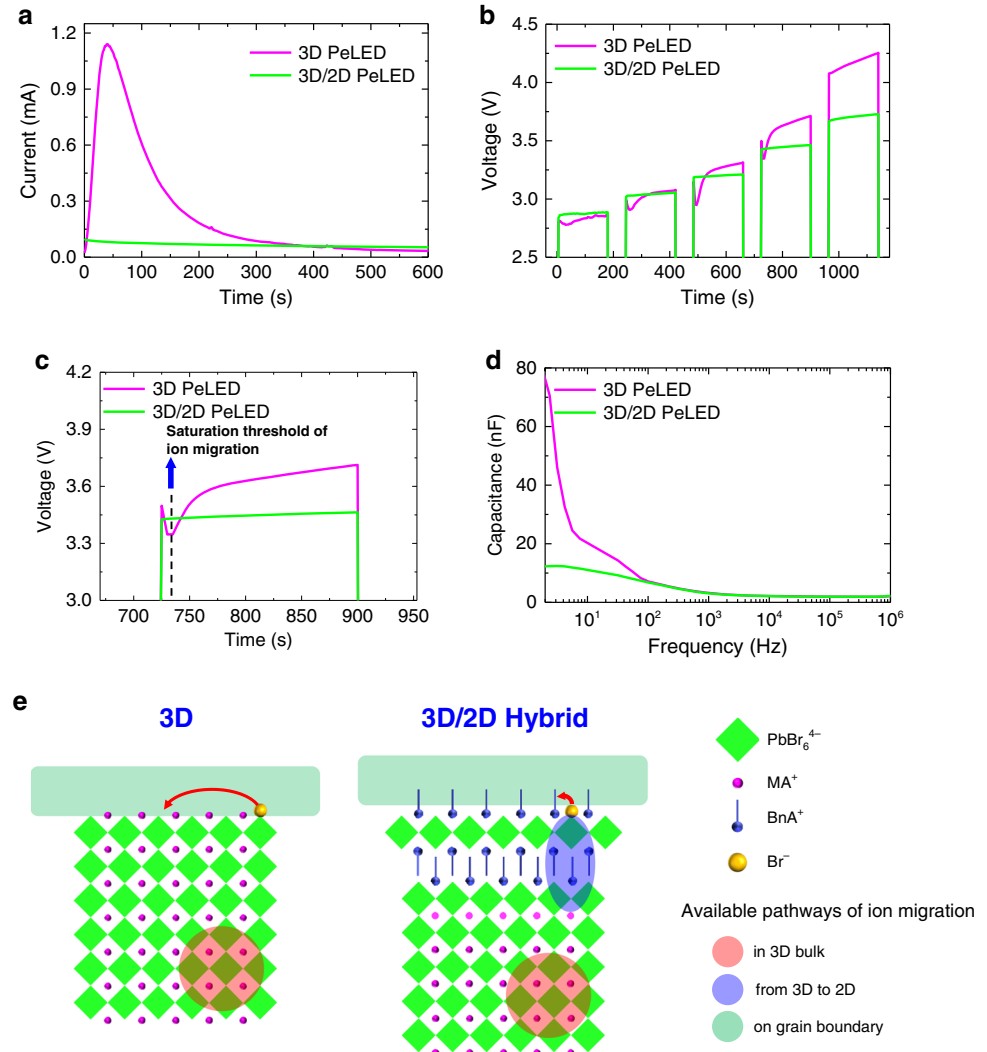

**Fig. 7 Influence of ion migration in perovskites on PeLEDs. a** Current behaviour of PeLEDs as a function of time under constant voltage of 3.5 V. **b** Voltage behaviour of PeLEDs under increasing step constant current of 0.16, 0.22, 0.44, 1.11, 2.22 mA cm⁻² from the left column to the right in a sequence. **c** Magnified version of one step showing the discrepancy in the voltage behaviour between 3D PeLED and 3D/2D hybrid PeLED. **d** Capacitance-frequency curve of PeLEDs. **e** Schematic of possible ion migration pathways in 3D and 3D/2D hybrid perovskites and efficient restriction of ion migration owing to the BnA which has retardation effect of reorientation.

examined this retardation effect more concretely by considering atomic-bond breaking associated with the organic-cation reorientation. In MAPbX₃, the two dominant hydrogen bonds should be broken to allow 180° reorientation of an MA⁺ adjacent to a migrating defect ion[46]. We defined the energy of the dominant hydrogen bond between a proton in NH₃ and halide ion in the PbX₆ octahedron unit as $E_{HB}$. Previous ab initio calculations based on quantum mechanics of atoms in molecule (QTAIM) obtained $E_{HB} \approx 200$ meV/MA⁺ for MAPbI₃[46,47]. In BnA⁺, the breaking of the π–π interaction themselves should be considered, in addition to the rupturing of this hydrogen bonding. According to ab initio calculations based on the density-functional tight-binding (DFTB) method, the energy $E_{\pi\pi}$ of the π–π stacking interaction between two neighbouring benzene rings is $-4.38$ kcal mol⁻¹[48]. For simplicity, we assume that $E_{HB}$ of BnA⁺ is approximately equal to that of MA⁺. This assumption does not alter our qualitative argument on the retardation effect caused by the substitution of BnA⁺ for MA⁺. We define the rate (frequency per unit time) of the reorientation of BnA⁺ and MA⁺ as $\nu_{BnA}$ and $\nu_{MA}$, respectively, and the retardation coefficient ($R_c$) as the ratio of these two frequencies.

According to the transition-state theory,

$$\nu_{BnA} = \frac{RT}{N_o h} e^{-(E_{HB}+2E_{\pi\pi})/(RT)}, \qquad (5)$$

where $N_o$ is Avogadro's number, $h$ is Planck's constant, and $R$ denotes the gas constant ($=1.987$ cal mol⁻¹ K⁻¹)[45]; Here, a factor of '2' is introduced in the exponent by considering that two π–π stackings must be broken for the reorientation of each BnA⁺ in the 3D/2D hybrid structure. Then the retardation coefficient can be written as $R_c \equiv \frac{\nu_{BnA}}{\nu_{MA}} \approx e^{-2E_{\pi\pi}/(RT)} = 4.23 \times 10^{-7}$ at 300 K. This is only an order-of-magnitude estimate, but it suggests strongly that the substitution of BnA⁺ for MA⁺ remarkably reduces the reorientation rate and, thereby significantly suppresses ion migration.

**Stability of PeLEDs and ion migration.** The 3D/2D hybrid PeLEDs also showed the excellent $I$–$V$–$L$ reliability under the repetition of sweeping voltage from 0 to 5 V and remarkably maintained its current density and luminance in spite of the repeating voltage sweep over 20 times (Fig. 8a–d). This

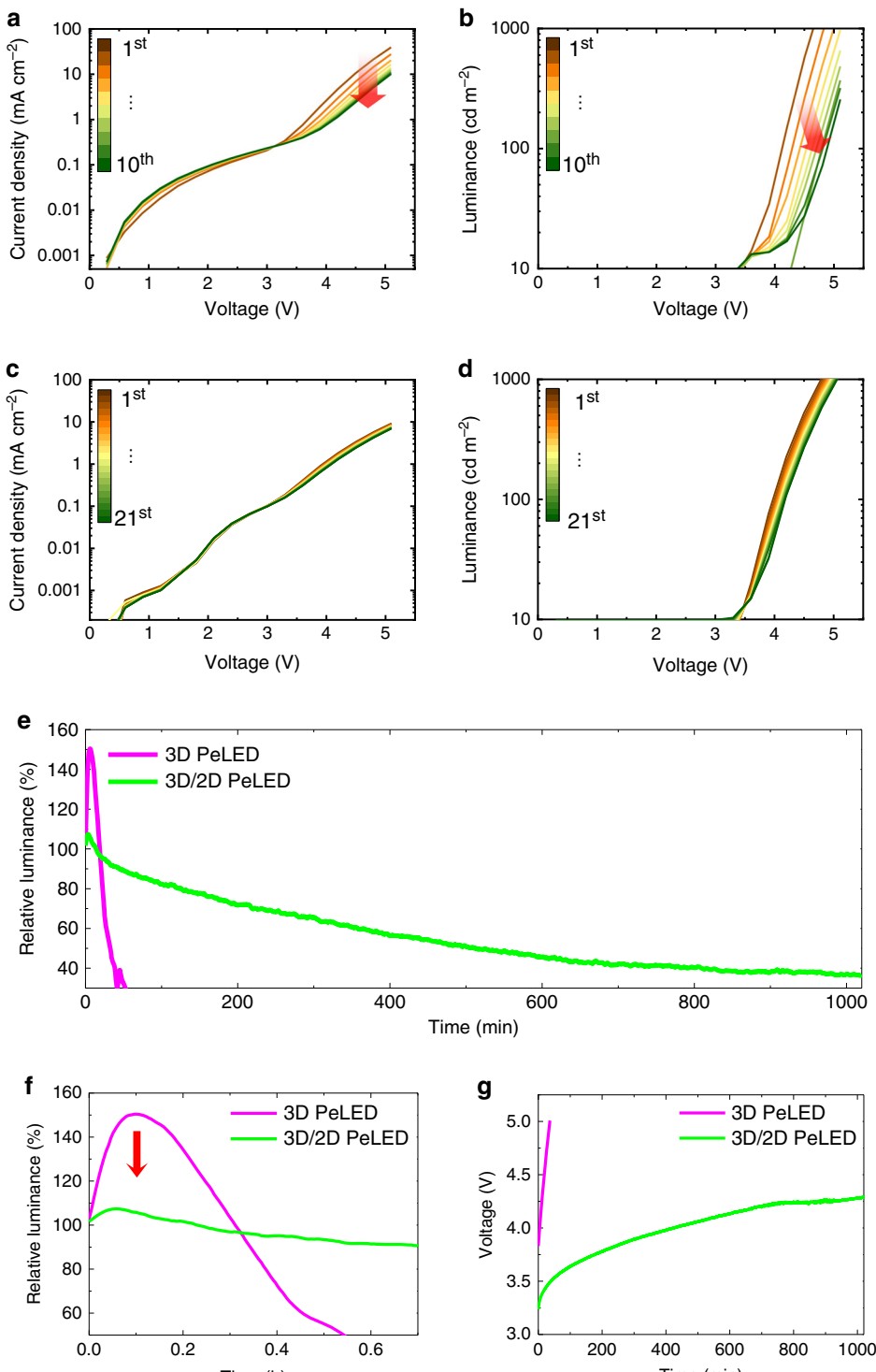

**Fig. 8 Reliability and long-term stability of PeLEDs. a** *J–V* and **b** *L–V* curves of 3D PeLEDs with a repetition of voltage sweep and **c**, **d** those of 3D/2D hybrid PeLED. **e** Relative luminance of 3D/2D hybrid PeLED (green) and of 3D PeLED (red) over time. **f** Magnified version of relative luminance of PeLEDs showing an extremely reduced overshoot of luminance. **g** Voltage curves of 3D/2D hybrid PeLED (green) and 3D PeLED (red) over time under constant current to make PeLEDs have an emission of 100 cd m$^{-2}$.

characteristic is a result of the suppressed ion migration. In contrast, the 3D PeLED showed degradation in current density and luminance in <10 such voltage sweeps. The 3D/2D hybrid PeLED also showed much less *J–V* hysteresis than the 3D PeLED under continuous forward and backward scans (Supplementary Fig. 17). This excellence of the stability demonstrates that the 3D/2D hybrid PeLED has much better operational stability than the

3D PeLED. The measurement was performed under CC condition for each device to elicit luminance of 100 cd m$^2$ (Fig. 8e). The luminance of the 3D PeLED overshot to 150.4% of the initial luminance, then dropped to 40% of the luminance after time $T_{40} = 38$ min. As ions migrate towards and accumulate at each end of the perovskite layer, the charge injection from the electrodes tends to be facilitated because of the strong local

electric field at the interfaces, and this facilitation results in the initial luminance overshoot. In contrast, the 3D/2D hybrid PeLED had a dramatically increased operating lifetime as $T_{40} = 810$ min, which is more than 21 times longer than $T_{40}$ of 3D PeLED. Moreover, the 3D/2D hybrid PeLED showed extremely reduced luminance overshoot as only 7.4% increase from the initial luminance, because the ion migration was effectively suppressed (Fig. 8f). We also investigated the operational stability of the PeLED using the ANI-added MAPbBr$_3$, which showed strong luminance overshoot as 146.7% compared to the initial luminance, implying that the addition of ANI cannot effectively suppress the ion migration (Supplementary Fig. 18). We summarized the luminance overshoot of PeLEDs from reported studies and compared their overshoot ratios (Supplementary Table 7 and Supplementary Fig. 19).

The applied voltage in the devices also showed a large difference as a function of time (Fig. 8g). The steeper voltage increase in the 3D PeLED indicates the inferior stability of the device compared to the 3D/2D hybrid PeLED. Also, accumulated ions at the interface can cause chemical corrosion of electrodes and thereby accelerate the degradation of the device[49]. The 3D/2D hybrid PeLED does not suffer from those problems because of the effectively suppressed ion migration resulting in the much longer operating lifetime than the 3D perovskite PeLED.

## Discussion

A perovskite light-emitter of proton-transfer-induced 3D/2D hybrid structure composed of dominant 3D MAPbBr$_3$ and a small amount of 2D BnA$_2$PbBr$_4$ has been developed by using a neutral amine reagent BnA instead of ammonium halide salt. A proton-transfer reaction between MA$^+$ and BnA occurs in the precursor state, and facilitates participation by BnA$^+$ in 2D perovskite crystallization. A 2D phase forms without destroying the 3D perovskite phase in the film. The incorporation of 2D perovskite into 3D perovskite restricts the formation of ionic defects because BnA$^+$ has a lower rotational degree of freedom than MA$^+$. Therefore, 3D/2D hybrid perovskite leads to efficient radiative recombination by deactivating even deep-level trap states. Also, the 3D/2D hybrid structure can suppress ion migration owing to the retardation effect in the reorientation of BnA$^+$ in the lattice based on π–π interaction of their aromatic rings by considering possible pathways of ion migration. The PeLEDs with the 3D/2D hybrid perovskite emitter show an extremely reduced initial luminance overshoot ratio as 7.4%, 21 times longer operational stability without noticeable overshoot of luminance, and significantly improved luminous efficiency. The material insights on 'proton-induced 3D/2D hybrid perovskites' we propose can help pave the way for the future practical applications of perovskite electronics (i.e. solar cells, LEDs, and memories) with a long operating lifetime.

## Methods

**Materials**. TPBi was purchased from OSM and MABr was purchased from Greatcell Solar. PbBr$_2$ (99.999%), ANI (ACS Reagent, 99.5%), BnA (ReagentPlus, 99%), LiF (99.99%, trace metal basis), DMSO (99.8%, anhydrous) were purchased from Sigma-Aldrich. All chemicals were used as received.

**Preparation of perovskite precursor solution**. MAPbBr$_3$ precursor solutions (35.0 wt%) were prepared by dissolving MABr and PbBr$_2$ in DMSO with a molar ratio of 1.06:1 (MABr:PbBr$_2$), and then stirred overnight. Before spin-coating the solution, the neutral amine additive (ANI or BnA) was added into the MAPbBr$_3$ precursor solution.

**Device fabrication**. For PeLEDs, the Buf-HIL layer was spin-coated on a cleaned FTO-coated glass at 3000 rpm for 90 s, and samples were annealed on a hot plate for 30 min at 150 °C in air. Buf-HIL solution was composed of PEDOT:PSS (Clevios P VP AI4083) and tetrafluoroethylene-perfluoro-3,6-dioxa-4-methyl-7-octene-sulfonic acid copolymer (PFI) (Sigma-Aldrich) (1:1 wt:wt). On the prepared

substrate, the prepared perovskite precursor solution was spin-coated at 3000 rpm, followed by the additive-based nanocrystal-pinning process to induce immediate crystallization of perovskite by using TPBi solution in chloroform[3]. The deposited perovskite film was annealed on a hot plate at 90 °C for 10 min. Subsequently, 50 nm of TPBI, 1 nm of LiF, and 100 nm of Al were thermally evaporated in sequence in a high-vacuum chamber to complete the device. For HODs, 30 nm of molybdenum oxide (MoO$_3$) and 50 nm of Au were thermally deposited on top of MAPbBr$_3$ to make a structure of glass/FTO/Buf-HIL/MAPbBr$_3$/MoO$_3$/Au in the high-vacuum chamber. The devices were encapsulated in N$_2$ atmosphere and the pixel area of a device was 4.8 mm$^2$.

**$^1$H NMR measurement**. The $^1$H NMR spectra were obtained using a high-resolution NMR spectrometer (Bruker Advance 600 MHz). For the measurement, the perovskite precursor solution was prepared as mentioned above, except that the precursors were dissolved in DMSO-$d_6$ (1 mL). For additive-mixed solutions, different amounts of BnA or ANI were added to the precursor solution as the experimental sample. The samples were prepared at room temperature in N$_2$ atmosphere, and their characteristics were measured at room temperature.

**$^1$H MAS NMR spectroscopy**. $^1$H MAS NMR spectra of the samples were collected at room temperature on a Bruker NMR system (14.1 T) at Larmor frequency of 600.41 MHz with a 1.9-mm triple-resonance Bruker NMR probe using a spin-echo pulse sequence at a spinning speed of 40 kHz. Fast sample spinning speed of 40 kHz yielded high-resolution $^1$H MAS NMR spectra. The spin-echo pulse sequence (π/2–t–π–t) was used to suppress $^1$H background signals from the probe[50,51]. The π/2 pulse length of 2 μs was applied for the samples and a recycle delay time of 10 s was used. Approximately 0.5–6 mg of perovskite powder after being scraped off the film was used. To improve the signal-to-noise ratio, around 16–256 scans were averaged in the $^1$H MAS NMR spectra. The $^1$H NMR spectra were referenced externally using tetramethylsilane (TMS) solution.

**Characterizations of perovskite films**. The absorbance of the perovskite film was measured using UV–vis absorption spectrophotometer (Cary-5000). Steady-state PL of the film was measured using a spectrofluorometer (JASCO FP6500). Xenon arc lamp with continuous output power of 150 W was used as an excitation source, and the excitation wavelength was 365 nm. The light was incident from the glass side. XRD analysis was carried out using X-ray diffractometer (PANalytical) with Cu kα radiation at a scan rate of 4°/min. GIXD was performed on the perovskite film at the 6D and 9A beamlines at the Pohang Accelerator Laboratory, Korea. Top-view and cross-sectional SEM images of the perovskite films were obtained using a field-emission SEM (MERLIN compact, ZEISS) at the Research Institute of Advanced Materials in Seoul National University. TCSPC measurement was conducted by using TCSPC module (FluoTime 300, PicoQuant). The instrument response function (IRF) was ignored because the PL lifetime curves were much longer than the temporal width of the IRF. The excitation power density was around 25 mW cm$^{-2}$, and the excitation wavelength was fixed at 405 nm. The excitation power density was determined by using a laser power meter (Thorlabs) for the excitation-dependent PL lifetime measurement. For the temperature-dependent PL measurement, the sample was mounted in a cryostat under vacuum condition. The target temperature was equilibrated by over 3 min stabilization before measuring a steady-state PL spectrum. UPS of the perovskite films was measured using a photoelectron spectrometer (Kratos Inc., AXIS-Ultra DLD) with He I radiation (21.2 eV) in collaboration with Korea Basic Science Institute (KBSI). PLQE of perovskite thin films on quartz substrate was measured by using the integrating sphere method with a continuous-wave 405 nm laser diode (2.5 mW cm$^{-2}$)[3].

**Transmission electron microscopy (TEM) analysis**. HAADF-STEM and energy-dispersive X-ray spectrum (EDXS) imaging were performed on an aberration-corrected Titan-Themis 60-300 (Thermo Scientific) at an acceleration voltage of 200 kV with probe currents of approximately 100 and 500 pA, respectively. This microscope is equipped with a high brightness Schottky X-FEG gun, a Super-X EDX system comprising four silicon drift detectors, and Velox acquisition software. The specimen for STEM analysis was prepared on a carbon support grid by dripping solution which was a dispersion of perovskite powder in chloroform after being scraped off the film.

**Characterization of PeLEDs**. The current–voltage–luminance characteristics of PeLEDs were measured using a source-measurement unit (Keithley 236), a spectroradiometer (Minolta CS-2000), and a control computer. The operational lifetime of the PeLEDs was measured using a lifetime measurement system (M6000, McScience) under a CC condition that yielded luminance of 100 cd m$^{-2}$. Capacitance–voltage, capacitance–frequency, and time-dependent $J$–$V$ characteristics were measured using a Solartron 1260 impedance/gain-phase analyzer with a Solartron 1287 potentiostat.

**Charge-density calculation details**. Charge-density calculations were conducted using a plane-wave pseudopotential approach within DFT as implemented in the

Viena Ab Initio Simulation Package[52]. The electron–core interactions were described with the projected-augmented wave pseudopotentials and the exchange and correlation functional were treated using the Perdew–Burke–Ernzerhof parametrization of GGA for structural relaxations[53,54]. The molecule was placed in the middle of a size-fixed cubic cell with $a = b = c = 30$ Å to optimize the atomic configuration. The electronic energy was minimized with a tolerance of $10^{-6}$ eV, and ionic relaxation was performed with a force tolerance of 0.01 eV Å$^{-1}$ on each ion. The kinetic energy cutoff was set as 520 eV, and a Gamma point was used for the Brillouin zone integration. The charge density along $b$-axis was integrated from the CHGCAR by $\rho(y) = \int \rho(x, y, z) dx\, dz$.

## Data availability

The data that support the findings of this study are available from the corresponding author upon reasonable request.

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

## Acknowledgements

This work was supported by the National Research Foundation of Korea (NRF) grant funded by the Korea government (Ministry of Science, ICT & Future Planning) (NRF-2016R1A3B1908431). N.-G.P. acknowledges financial support from the National Research Foundation of Korea (NRF) grants funded by the Ministry of Science and ICT (MSIT) of Korea under contracts NRF-2014M3A6A7060583 (Global Frontier R&D Programme on Center for Multiscale Energy System). The $^1$H MAS NMR work was supported by the National Research Foundation of Korea (NRF) grant funded by the Ministry of Science and ICT to S.K. Lee (2017R1A2A1A17069511). A.S., S.N. and R.H.F. acknowledge funding and support from the SUNRISE project (EP/P032591/1), EPSRC and UKIERI project. S.N. would like to acknowledge funding and support from the Royal Society Newton International Fellowship.

## Author contributions

H.K. and J.S.K. conceptualized the study. H.K., J.S.K., and J.-M.H. developed devices and performed measurements. Z.L., L.Z., and H.M.J. applied computational technique. H.K., J.S.K., and J.-M.H., M.P., I.-H.P., H.J.Y., M.-H.P., S.-H.J., Y.-H.K., J.-W.P., E.O., S.N., A.S., J.J.K., S.K.L., H.Y., K.P.L., N.-G.P. conducted the research and investigation process. H.K. and J.S.K., wrote the initial draft. H.K., J.S.K., J.-M.H., L.Z., S.K.L., H.Y., H.M.J., R.H.F., K.P.L., M.K.N., N.-G.P., and T.-W.L. reviewed and edited the paper. T.-W.L. supervised.

## Competing interests

The authors declare no competing interests.
