## [Peer Review File · Nature Communications]

Reviewers' comments:

Reviewer #1 (Remarks to the Author):

In this work, Kim et al. fabricated 3D/2D hybrid perovskites by introducing neutral benzylamine. It is reported that the ion migration is suppressed efficiently. With this method, the authors reduced luminance overshoot well and achieved good device stability. In fact, the points discussed in this manuscript are the most concerned things in this field while present results might not meet the high quality of this journal. Following are my questions and advices which may help to improve the manuscript.

minor comments:

1. There are too many figures in the main text, and some of them are not necessary in my opinion. There are also some mistakes in the ms. So the authors should polish it carefully.
2. The XRD results are very strange, the diffraction peaks are so wide, and even there are many small peaks in one peak.
3. For a paper focusing on stability, the present device efficiency (EQE) is too low.

major comments:

1. For the PL improvement, the authors gave many results to support the claim of defect passivation, however, these measurements are just from the aspect of final states. How can the 2D part passivate 3D part? Is there any structure model with solid evidence? Besides, the authors measured UPS results to claim the quantum well structure, it is not logically. What we want see is good evidence. Why BnA-30% shows weaker PL intensity compared to BnA-50%?
2. For the suppression of ion migration, the mechanism is mainly based on theoretical calculation, it is not enough. Since the direction of ion migration is along the direction of electric field, can the authors give a clear 3D/2D structure distribution? with this result, the ms will be well and many of concerns will be answered.
3. Can the authors prepare 3D/2D structure with protonated BnA and MA?

Reviewer #2 (Remarks to the Author):

The authors report on the improvement of characteristics (increase in luminous efficiency, reduced overshoot) and operation lifetime of the hybrid 3D/2D perovskite light-emitting diodes. The improvements were achieved via the introduction of neutral benzylamine (BnA) to methylammonium lead bromide perovskites, and as the author claim, it helped to reduce the formation of deep-traps and ion-migration due to trap-state passivation of 3D perovskite with the 2D perovskite. I have to emphasize that the study is interesting and presented in a well-organized and logical way. In this study, the authors used many experimental techniques that were supplemented with simulation results; this allowed for detailed exploration chemistry of the hybrid 3D/2D perovskite formation, ion migration, charge carrier transport and other essential events in both films and 3D/2D hybrid perovskite LEDs. Given the high quality of the work, I recommend publication of this paper in Nature Communications after addressing the following comments:

1. In addition to benzylamine, the authors also used aniline (ANI) for the formation of the perovskite. But the reasons behind which this molecule was used are not entirely clear; thus, I suggest to provide more details on why ANI was used, this also will give more clarity for the reader.
2. Figure 3 e shows that the structure of 3D/2D perovskite containing BnA is more grained compared to pristine MAPbBr₃ and MAPbBr₃ with 2.4 mol% of ANI. The authors claim that the presence of small

grains can be beneficial for the efficient exciton formation within a grain and provides relevant references. Authors should note that based on more recent studies (10.1021/acsnano.6b02734) exciton binding energies of MAPbBr₃ were found to be similar or even below thermal energies at room temperature, and it is likely that free charges are formed instead of excitons. Overall, the benefit of obtaining grained morphology seems to me not very desirable.

3. Figure 4 shows that the PL intensity changes depending on the perovskite composition and additive concentrations. It is evident that the PL intensity is very sensitive on the position of the sample which makes it difficult to measure its intensity accurately; thus for the accurate estimation of the intensity differences, PL Quantum yield measurements would be more useful.

4. The hole-only devices employed molybdenum oxide layer on top of the perovskite. However, information on how these devices were made and MoO₃ layer was deposited is missing.

Minor issues:

- Please, define what is Buf-HIL when first time used in the text.
- Line 201. Unknown character after the "type-".
- Some references (5; 14; 26; 46;) should be corrected since complete bibliographic information is missing.

Reviewer #3 (Remarks to the Author):

The authors report on the role of 3D/2D hybrid perovskite materials in suppressing ion migration, which, as a result, reduces the overshoot of luminance over time as compared to 3D MAPbBr₃ perovskites in perovskite LEDs. This was achieved by adding neutral benzylamine that was assumed to induce proton transfer from MA to facilitate the crystallization of the 2D perovskite, which is, as a result, claimed to suppress the formation of trap states and ion migration, enhancing operational stability and efficiency. While the study presented is very insightful in general and impactful for the community, a number of claims have not been rigorously assessed and remain speculative, such as the role of proton transfer in the entire process, although highlighted throughout the manuscript. For this purpose, this reviewer recommends a major revision of the study before considering further for publication. Some of the critical remarks that should be addressed are the following.

- The authors highlight the role of proton-transfer in the analyzed properties without providing direct evidence that such a process is instrumental in the solid-state. Specifically, although solution-based NMR spectra provide valuable insights, this does not provide evidence in the solid-state. Moreover, it remains unclear whether the proton transfer is a critical factor determining the properties or the usage of benzylamine leads to the observed effects. For this purpose, molecular dynamics simulations in conjunction with control experiments employing benzylammonium salts would be helpful. Importantly, the effects should be described without excluding other contributing factors in the analysis.

- The authors should clarify in the main manuscript whether monitoring the protonation reaction via UV-vis spectroscopy was performed in solution or in the corresponding thin-films.

- The study concludes that the higher propensity of the benzylamine to form the 2D perovskite phase as compared to the aniline is based on the difference in their corresponding pK_a. This is, however, rather misleading, as the geometry of the molecules also defines the ability to effectively form layered 2D phases. In this regard, the presence of the methylene linker enables effective and adaptable

penetration into the A-cation vacancy for the benzylammonium system, unlike the anilinium, which would result in the formation of a better-defined layered structure. In fact, anilinium-based 2D phase could co-exist, yet its poor orientation/crystallinity would prevent it from being observed experimentally via X-ray diffraction. Considering these factors, the authors are strongly advised not to exclusively ascribe the observed properties to the proton transfer effects. This is particularly misleading in the title of the manuscript.

- The reduced reorientation rate of the benzylammonium system is ascribed to the π - π interactions of the neighboring rings. In addition, however, the authors should also consider the steric repulsion of the molecules that should be mentioned.

- The analysis of the effect of the additives on the optoelectronic properties focuses on the comparison of the benzylamine-containing system and the pristine perovskite. However, the authors should also perform a control study involving aniline-containing perovskite materials, as, despite the lack of apparent effect on the structural properties, their presence could suppress ion migration and affect the performance.

- The literature mostly appears appropriate. However, the work of Loi et al. (*Appl. Phys. Rev.* 2019, 6, 031401), for instance, has not been cited, despite reporting on the relevant usage of benzylamine in hybrid perovskites. In addition, the role of adaptability through the introduction of a flexible linker in the formation of layered perovskite structures should be referred to (for instance, general role of the linker: *CrystEngComm*, 2010, 12, 2646 and the recent methylene group case: *Adv. Energy Mater.* 2019, 1900284).

Thank you for considering the aforementioned remarks, which I believe would contribute to enhancing the impact of this interesting study.

Answers to Reviewers

First of all, we would like to appreciate the reviewer's valuable comments. We answered the reviewer's questions as below. We also revised our manuscript to comply with the reviewer's comments.

Reviewer #1 (Remarks to the Author):

In this work, Kim et al. fabricated 3D/2D hybrid perovskites by introducing neutral benzylamine. It is reported that the ion migration is suppressed efficiently. With this method, the authors reduced luminance overshoot well and achieved good device stability. In fact, the points discussed in this manuscript are the most concerned things in this field while present results might not meet the high quality of this journal. Following are my questions and advices which may help to improve the manuscript.

Minor comments:

1. There are too many figures in the main text, and some of them are not necessary in my opinion. There are also some mistakes in the ms. So the authors should polish it carefully.

Answer)

We appreciate the reviewer's valuable comment. According to the Reviewer's suggestion, we shifted Fig. 3a showing the grazing-incident X-ray diffraction (GIXD) 1D profile to the Supplementary information. We also corrected some mistakes and typos in the manuscript.

Revised parts in the manuscript)

In Fig. 3,

Fig. 3 Crystal structure and morphology of MAPbBr₃ according to the addition of ANI or BnA. **a-c** 2D GIXD patterns of pristine MAPbBr₃, MAPbBr₃ with 2.4 mol% ANI, and 2.4 mol% BnA. **d** Cross-sectional SEM images of pristine MAPbBr₃ (top), MAPbBr₃ with 2.4 mol% ANI (middle) and 2.4 mol% BnA (bottom).

In the page 7, line 13,

The information about the crystal structure of perovskite also can be shown in 2D GIXD patterns (Fig. 3a-c).

In the page 8, line 3,

The films of pristine MAPbBr₃ and MAPbBr₃ with 2.4 mol% ANI had the similar columnar grain structure, whereas 3D/2D hybrid perovskite film with 2.4 mol% BnA had granular grains with a smaller size (Fig. 3d).

In the Supplementary Figure 3.

Supplementary Figure 3 (a-g) 2D GIXD patterns of pristine MAPbBr₃ film (a), with ANI 2.4 mol% (b), ANI 15 mol% (c), ANI 30 mol% (d), BnA 2.4 mol% (e), BnA 15 mol% (f), and BnA 30 mol% (g). 1D XRD profiles of pristine MAPbBr₃ film, MAPbBr₃ with ANI 2.4 mol% and BnA 2.4 mol% (h). 1D XRD profiles of MAPbBr₃ with different amounts of ANI (i) and BnA (j). Traces have been offset vertically for clarity.

In the page 7, line 8.

In contrast, the addition of 2.4 mol% BnA led to an appearance of additional peaks at low angles; the most dominant one was at $2\theta = 5.32^\circ$ (Supplementary Figure 3).

2. The XRD results are very strange, the diffraction peaks are so wide, and even there are many small peaks in one peak.

Answer)

We appreciate the reviewer's valuable comment. In a 2D pattern of GIXD, thin films including randomly-oriented crystal grains show Debye ring-like patterns along a given scattering vector value, originating from the randomly-oriented crystal planes with the same

layer spacing. As perovskite grains have high crystallinity and ordering, highly intense spike peaks are indicated in the 2D GIXD pattern (Fig. 3 in the revised manuscript). Therefore, some photo-diodes with limited photon counts in a CCD detector become early saturated, while others can still count the scattered photons during the exposure time. As a result, 1D X-ray profiles which were extracted along 45° from Q_z axis in the 2D GIXD patterns contained plateau regions by the saturated photo-diodes (Supplementary Figure 2h). However, the plateau peaks have actually high intensities. This is a common phenomenon when the X-ray analysis is performed on a specimen including single crystal-like grains. It is also possible to minimize the spike-like peaks via short-time exposure, but the other X-ray reflections will be decreased or disappeared.

3. For a paper focusing on stability, the present device efficiency (EQE) is too low.

Answer)

We appreciate the reviewer's valuable comment. Even though the EQE is not super high, we still believe that our work provides considerable insights regarding how to overcome ion migration problem in halide perovskites and perovskite light-emitting diodes (PeLEDs), and also provides a scientific guideline to improve stability of PeLEDs which is an important prerequisite for the future practical applications. We firmly believe that our work will expand the field of PeLEDs and stimulate considerable research on fundamental aspects of stability of PeLEDs and our approach in PeLEDs will be effectively employed in other various perovskite electronics such as solar cells, photodetectors, and memories.

Major comments:

1-1. For the PL improvement, the authors gave many results to support the claim of defect passivation, however, these measurements are just from the aspect of final states. How can the 2D part passivate 3D part? Is there any structure model with solid evidence?

Answer)

We appreciate the reviewer's incisive comment. As the reviewer mentioned, we presented many results to support the claim of defect passivation and briefly described how the 2D perovskite can passivate the 3D perovskite as *"In addition, the 2D perovskite reduces the defect density by passivating dangling bonds on 3D perovskite grains, which can act as*

charge trap sites leading to the increased rate of non-radiative recombination.” (see page 9, line 20 in the revised manuscript). We mainly attributed the improvement in PL characteristics of the 3D/2D hybrid perovskite to passivation of deep level traps which are associated with ionic defects because the incorporation of 2D perovskite can suppress the formation and migration of the ionic defects thereby minimizing a loss of charge carrier from quenching by the deep traps.

The 2D perovskite has a structure in which the ammonium terminal of the protonated benzylamine (BnA^+) interacts with the surface terminal $[\text{PbBr}_6]^{4-}$ octahedra of the 3D perovskite via hydrogen bonding.^{R1,R2} While MA^+ in the 3D perovskite can relieve steric hindrance and allow to accommodate extra defect such as interstitial bromide in the lattice,^{R3,R4} the presence of BnA^+ in the 2D perovskite lattice can effectively restrict the defect formation. Also, the following ion migration can be suppressed due to their intermolecular π - π interaction and steric repulsion associated with its low rotational degree of freedom in the lattice.

In order to scrutinize the structure of the 3D/2D hybrid perovskites in grain scale, we used high-angle annular dark-field scanning transmission electron microscopy (HAADF-STEM) (Fig. R1a). The specimen was prepared on a carbon support TEM grid by dripping solution which was a dispersion of perovskite powder in chloroform after being scraped from the film. It is challenging to obtain an image of the 3D perovskite domains due to an instant degradation by the strong electron beam energy and the unstable nature of the 3D perovskite.^{R5-R9} Nevertheless, we could clearly observe the 2D perovskite layer with $n = 1$ (Fig. R1b). It is worth noting that the smaller d -spacing (1.15 nm) than that obtained by grazing-incident X-ray diffraction (GIXD) analysis (1.66 nm) can be attributed to transient lattice contraction due to the high beam energy.^{R10,R11} Interestingly, the periodic 2D perovskite was only observable at the shell region of the grain. Because the 2D perovskite has lower surface energy than the 3D perovskite due to its fewer surface dangling bonds and surface relaxation, growth of the 2D perovskite most possibly occurs on the 3D perovskite grains and it also makes an interface with grain boundary which has the highest surface energy, thereby lowering the total potential energy of the system.^{R10,R12} Therefore, the mounted 2D perovskite on the 3D perovskite can effectively passivate the traps and block ion migration. We also monitored the composition of the perovskite grains using energy dispersive X-ray spectrometry in scanning TEM mode (EDXS-STEM) (Fig. R1c-f). HAADF-STEM images and corresponding EDXS elemental maps indicate that the grain is composed

of Pb and Br elements. We added the result of the TEM analysis with a corresponding description in the revised manuscript.

Fig. R1 (a) HAADF scanning TEM image of a 3D/2D hybrid perovskite grain. (b) High-resolution image of the grain framed with the yellow box in (a) showing the 2D perovskite ($n = 1$) with d -spacing of 1.15 nm. (c) A lower magnification HAADF scanning TEM image of the 3D/2D hybrid perovskite grain shown in (a), corresponding EDXS elemental maps of (d) Pb and (e) Br, and (f) integrated EDX spectrum of the acquired dataset.

- R1. Li, X. *et al.* Improved performance and stability of perovskite solar cells by crystal crosslinking with alkylphosphonic acid ω -ammonium chlorides. *Nature Chemistry* **7**, 703–711 (2015).
- R2. Ming Koh, T. *et al.* Enhancing moisture tolerance in efficient hybrid 3D/2D perovskite photovoltaics. *Journal of Materials Chemistry A* **6**, 2122–2128 (2018).
- R3. Meggiolaro, D., Mosconi, E. & De Angelis, F. Formation of Surface Defects Dominates Ion Migration in Lead-Halide Perovskites. *ACS Energy Lett.* **4**, 779–785 (2019).
- R4. Mosconi, E. & De Angelis, F. Mobile Ions in Organohalide Perovskites: Interplay of Electronic Structure and Dynamics. *ACS Energy Lett.* **1**, 182–188 (2016).

- R5. Li, Y. *et al.* Unravelling Degradation Mechanisms and Atomic Structure of Organic-Inorganic Halide Perovskites by Cryo-EM. *Joule* **3**, 2854–2866 (2019).
- R6. Rothmann, M. U. *et al.* Structural and Chemical Changes to $\text{CH}_3\text{NH}_3\text{PbI}_3$ Induced by Electron and Gallium Ion Beams. *Adv. Mater.* **30**, 1800629 (2018).
- R7. Yang, B. *et al.* Observation of Nanoscale Morphological and Structural Degradation in Perovskite Solar Cells by in Situ TEM. *ACS Appl. Mater. Interfaces* **8**, 32333–32340 (2016).
- R8. Dang, Z. *et al.* Low-Temperature Electron Beam-Induced Transformations of Cesium Lead Halide Perovskite Nanocrystals. *ACS Omega* **2**, 5660–5665 (2017).
- R9. Chen, X. & Wang, Z. Investigating chemical and structural instabilities of lead halide perovskite induced by electron beam irradiation. *Micron* **116**, 73–79 (2019).
- R10. Ferguson, K. R. *et al.* Transient lattice contraction in the solid-to-plasma transition. *Science Advances* **2**, e1500837 (2016).
- R11. Maehlen, J. P., Mongstad, T. T., You, C. C. & Karazhanov, S. Lattice contraction in photochromic yttrium hydride. *Journal of Alloys and Compounds* **580**, S119–S121 (2013).
- R12. Yang, Y., Gao, F., Gao, S. & Wei, S.-H. Origin of the stability of two-dimensional perovskites: a first-principles study. *Journal of Materials Chemistry A* **6**, 14949–14955 (2018).

Revised parts in the manuscript

In the Supplementary Figure 15,

Supplementary Figure 15 (a) HAADF scanning TEM image of a 3D/2D hybrid perovskite grain. (b) High-resolution image of the grain framed with the yellow box in (a) showing the 2D perovskite ($n = 1$) with d -spacing of 1.15 nm. (c) A lower magnification HAADF scanning TEM image of the 3D/2D hybrid perovskite grain shown in (a), corresponding EDXS elemental maps of (d) Pb and (e) Br, and (f) integrated EDX spectrum of the acquired dataset.

In order to scrutinize the structure of the 3D/2D hybrid perovskites in grain scale, we used high-angle annular dark-field scanning transmission electron microscopy (HAADF-STEM) and clearly observed the 2D perovskite layer with $n = 1$ (Supplementary Figure 15a, b). It is worth noting that the smaller d -spacing (1.15 nm) than that obtained by grazing-incident X-ray diffraction (GIXD) analysis (1.66 nm) can be attributed to transient lattice contraction due to the highly strong beam intensity.^{48,49} Interestingly, the highly periodic 2D perovskite was only observable at the shell region of the grain. Because the 2D perovskite has lower surface energy than the 3D perovskite due to its fewer surface dangling bonds and surface relaxation, growth of the 2D perovskite most possibly occurs on the 3D perovskite grains and it also

makes an interface with grain boundary which has the highest surface energy, thereby lowering the total potential energy of the system.^{48,50} Therefore, the mounted 2D perovskite on the 3D perovskite can effectively passivate the traps and block ion migration. We also monitored the composition of the perovskite grains using energy dispersive X-ray spectrometry in scanning TEM mode (EDXS-STEM). HAADF-STEM images and corresponding EDXS elemental maps indicate that the grain is composed of Pb and Br elements (Supplementary Figure 15c-f).

In the Methods,

Transmission electron microscopy (TEM) analysis. High-angle annular dark-field scanning transmission electron microscopy (HAADF-STEM) and energy-dispersive X-ray spectrum (EDXS) imaging were performed on an aberration-corrected Titan-Themis 60-300 (Thermo Scientific) at an acceleration voltage of 200 kV with probe currents of approximately 100 pA and 500 pA, respectively. This microscope is equipped with a high brightness Schottky X-FEG gun, a Super-X EDX system comprising four silicon drift detectors, and Velox acquisition software. The specimen for STEM analysis was prepared on a carbon support grid by dripping solution which was a dispersion of perovskite powder in chloroform after being scraped from the film.

In the Reference,

48. Ferguson, K. R. *et al.* Transient lattice contraction in the solid-to-plasma transition. *Science Advances* **2**, e1500837 (2016).
49. Maehlen, J. P., Mongstad, T. T., You, C. C. & Karazhanov, S. Lattice contraction in photochromic yttrium hydride. *Journal of Alloys and Compounds* **580**, S119–S121 (2013).
50. Yang, Y., Gao, F., Gao, S. & Wei, S.-H. Origin of the stability of two-dimensional perovskites: a first-principles study. *Journal of Materials Chemistry A* **6**, 14949–14955 (2018).

1-2. Besides, the authors measured UPS results to claim the quantum well structure, it is not logically. What we want see is good evidence.

Answer)

We observed the presence of phase pure 2D perovskite mostly with $n = 1$ in the 3D/2D hybrid perovskite by various characterizations such as absorbance, photoluminescence,

and GIXD. This is different from quasi-2D perovskite with lower phase purity in which various phases can concurrently exist according to its composition. Thus, we measured ultraviolet photoemission spectroscopy (UPS) of each 3D MAPbBr₃ and 2D BnA₂PbBr₄ film separately because one cannot measure the energy level at the interface between 3D and 2D perovskites where there might be a distortion or a misalignment of lattice at the junction.

1-3. Why BnA-30% shows weaker PL intensity compared to BnA-50%?

Answer)

We appreciate the reviewer's valuable comment. To confirm the steady-state photoluminescence (PL) result, we additionally performed the absolute PL quantum efficiency (PLQE) measurement (Fig. R2). The PLQE result showed a similar trend to the steady-state PL result according to the amount of BnA. When the small amount of BnA (2.4 mol%) was added into MAPbBr₃, the film showed the highest PL intensity and PLQE, and a further increase of BnA up to 30 mol% led to a decrease in the PL intensity and the PLQE. However, when a large amount of BnA (50 mol%) was added, the PL intensity and the PLQE of the film increased again. We may attribute this behaviour to a phase transition of perovskite from 3D/2D hybrid phase to quasi-2D phase as the amount of BnA increases. To be specific, the increase in the amount of BnA in the 3D/2D hybrid phase regime did not cause a significant shift of PL peak position because the 3D perovskite can maintain its crystalline structure. However, the increasing amount of 2D perovskite can relatively decrease the amount of 3D perovskite reducing the PL intensity of 3D perovskite and PLQE. In the quasi-2D regime, on the other hand, an increase in the amount of BnA caused a significant blue shift of PL peak position to 536 nm for 30 mol% and 530 nm for 50 mol%. The recovered PL intensity and the PLQE in the quasi-2D regime at 50 mol% can be attributed to the optimized energy funnelling and confinement effect in the quasi-2D perovskite structure.^{R13} The relatively lower PLQE compared to our previous report can be attributed to the use of low-power laser diode (1.98 mW) due to the limitation of access to the previous set-up.^{R14} We added a brief description of the phase transition and included the PLQE result in the manuscript to support and confirm the steady-state PL result.

Fig. R2 (a) PL spectra, (b) PL peak position and (c) PLQE of MAPbBr₃ film with different amounts of BnA.

R13. Yang, X. *et al.* Efficient green light-emitting diodes based on quasi-two-dimensional composition and phase engineered perovskite with surface passivation. *Nature Communications* **9**, 570 (2018).

R14. Cho, H. *et al.* Overcoming the electroluminescence efficiency limitations of perovskite light-emitting diodes. *Science* **350**, 1222–1225 (2015).

Revised parts in the manuscript)

In the page 9, line 3,

Only the addition of the larger amount of BnA caused a blue-shift of PL emission (to 537 nm for 30 mol% and to 530 nm for 50 mol%), accompanied by the appearance of 2D perovskite PL emissions at 405 nm and 435 nm, which can be assigned to $n = 1$ and 2, respectively. The further addition of BnA up to 100 mol% which corresponds to the same amount as MA, caused a dramatic increase in the intensity of PL emission at 409 nm from 2D perovskite. This shift can be more clearly seen in normalized PL spectra (Fig. 4d). **We also measured the absolute PL quantum efficiency (PLQE) which showed a similar trend to the steady-state PL intensity according to the amount of BnA (Supplementary Figure 5).** In contrast, however, increasing the amount of ANI only decreased the PL intensity, and did not shift the emission wavelength, which can be attributed to intensive non-radiative recombination due to the electron deficiency of ANI (Supplementary Figure 6).³⁰

In the Supplementary Figure 5,

Supplementary Figure 5 PLQE of MAPbBr₃ film according to the amount of BnA addition. The films were excited with a continuous-wave 405 nm laser diode (2.5 mW cm⁻²).

In the Methods,

PLQE of perovskite thin films on a quartz substrate was measured by using the same integrating sphere method reported previously with a continuous-wave 405 nm laser diode (2.5 mW cm⁻²).

2. For the suppression of ion migration, the mechanism is mainly based on theoretical calculation, it is not enough. Since the direct of ion migration along the direct of electric field, can the authors give a clear 3D/2D structure distribution? with this result, the ms will be well and many of concerns will be answered.

Answer)

We appreciate the reviewer’s constructive comment. Actually, we already presented many experimental clues that can prove the suppression of ion migration by using 3D/2D hybrid perovskite in Figure 7, 8. The reason why the 3D/2D hybrid perovskite can more effectively suppress the ion migration than the 3D perovskite can be mainly attributed to the granular structure formation and 2D shell formation in 3D/2D hybrid perovskites. Scanning electron microscopy (SEM) images showed that the 3D/2D hybrid perovskite film had granular-like grains with a smaller size while the pristine MAPbBr₃ had the columnar-like grain structure (Fig. 3e in the revised manuscript). As the ion migration mostly occurs on grain boundaries (GBs), mobile ions have to migrate much longer pathway in the granular-

like 3D/2D hybrid perovskite than in the columnar-like 3D perovskite (Fig. R1). Furthermore, the 3D/2D hybrid perovskite has a much greater number of boundary nodes where migrating ions along the direction of the external electric field can be blocked and lose their kinetic energy leading to the termination of their migration.

Fig. R3 Schematic showing ion migration pathway in 3D (left) and 3D/2D hybrid (right) perovskite.

In order to scrutinize the structure of the 3D/2D hybrid perovskites in grain scale, we used high-angle annular dark-field scanning transmission electron microscopy (HAADF-STEM) (Fig. R1a). The specimen was prepared on a carbon support TEM grid by dripping solution which was a dispersion of perovskite powder in chloroform after being scraped from the film. It is challenging to obtain an image of the 3D perovskite domains due to an instant degradation by the strong electron beam energy and the unstable nature of the 3D perovskite.^{R5-R9} Nevertheless, we could clearly observe the 2D perovskite layer with $n = 1$ (Fig. R1b). It is worth noting that the smaller d -spacing (1.15 nm) than that obtained by grazing-incident X-ray diffraction (GIXD) analysis (1.66 nm) can be attributed to transient lattice contraction due to the high beam energy.^{R10,R11} Interestingly, the periodic 2D perovskite was only observable at the shell region of the grain. Because the 2D perovskite has lower surface energy than the 3D perovskite due to its fewer surface dangling bonds and surface relaxation, growth of the 2D perovskite most possibly occurs on the 3D perovskite grains and it also makes an interface with grain boundary which has the highest surface energy, thereby lowering the total potential energy of the system.^{R10,R12} Therefore, the mounted 2D perovskite on the 3D perovskite can effectively passivate the traps and block ion migration. We also monitored the composition of the perovskite grains using energy

dispersive X-ray spectrometry in scanning TEM mode (EDXS-STEM) (Fig. R1c-f). HAADF-STEM images and corresponding EDXS elemental maps indicate that the grain is composed of Pb and Br elements. We added the result of the TEM analysis with a corresponding description in the revised manuscript.

Fig. R1 (a) HAADF scanning TEM image of a 3D/2D hybrid perovskite grain. (b) High-resolution image of the grain framed with the yellow box in (a) showing the 2D perovskite ($n = 1$) with d -spacing of 1.15 nm. (c) A lower magnification HAADF scanning TEM image of the 3D/2D hybrid perovskite grain shown in (a), corresponding EDXS elemental maps of (d) Pb and (e) Br, and (f) integrated EDX spectrum of the acquired dataset.

- R1. Li, X. *et al.* Improved performance and stability of perovskite solar cells by crystal crosslinking with alkylphosphonic acid ω -ammonium chlorides. *Nature Chemistry* **7**, 703–711 (2015).
- R2. Ming Koh, T. *et al.* Enhancing moisture tolerance in efficient hybrid 3D/2D perovskite photovoltaics. *Journal of Materials Chemistry A* **6**, 2122–2128 (2018).
- R3. Meggiolaro, D., Mosconi, E. & De Angelis, F. Formation of Surface Defects Dominates Ion Migration in Lead-Halide Perovskites. *ACS Energy Lett.* **4**, 779–785 (2019).

- R4. Mosconi, E. & De Angelis, F. Mobile Ions in Organohalide Perovskites: Interplay of Electronic Structure and Dynamics. *ACS Energy Lett.* **1**, 182–188 (2016).
- R5. Li, Y. *et al.* Unravelling Degradation Mechanisms and Atomic Structure of Organic-Inorganic Halide Perovskites by Cryo-EM. *Joule* **3**, 2854–2866 (2019).
- R6. Rothmann, M. U. *et al.* Structural and Chemical Changes to CH₃NH₃PbI₃ Induced by Electron and Gallium Ion Beams. *Adv. Mater.* **30**, 1800629 (2018).
- R7. Yang, B. *et al.* Observation of Nanoscale Morphological and Structural Degradation in Perovskite Solar Cells by in Situ TEM. *ACS Appl. Mater. Interfaces* **8**, 32333–32340 (2016).
- R8. Dang, Z. *et al.* Low-Temperature Electron Beam-Induced Transformations of Cesium Lead Halide Perovskite Nanocrystals. *ACS Omega* **2**, 5660–5665 (2017).
- R9. Chen, X. & Wang, Z. Investigating chemical and structural instabilities of lead halide perovskite induced by electron beam irradiation. *Micron* **116**, 73–79 (2019).
- R10. Ferguson, K. R. *et al.* Transient lattice contraction in the solid-to-plasma transition. *Science Advances* **2**, e1500837 (2016).
- R11. Maehlen, J. P., Mongstad, T. T., You, C. C. & Karazhanov, S. Lattice contraction in photochromic yttrium hydride. *Journal of Alloys and Compounds* **580**, S119–S121 (2013).
- R12. Yang, Y., Gao, F., Gao, S. & Wei, S.-H. Origin of the stability of two-dimensional perovskites: a first-principles study. *Journal of Materials Chemistry A* **6**, 14949–14955 (2018).

Revised parts in the manuscript)

In the Supplementary Figure 14,

Supplementary Figure 14 Schematic showing ion migration pathway in 3D (left) and 3D/2D hybrid (right) perovskite.

In the Supplementary Figure 15,

Supplementary Figure 15 (a) HAADF scanning TEM image of a 3D/2D hybrid perovskite grain. (b) High-resolution image of the grain framed with the yellow box in (a) showing the 2D perovskite ($n = 1$) with d -spacing of 1.15 nm. (c) A lower magnification HAADF scanning TEM image of the 3D/2D hybrid perovskite grain shown in (a), corresponding

EDXS elemental maps of (d) Pb and (e) Br, and (f) integrated EDX spectrum of the acquired dataset.

In the page 15, line 18,

Also, the 3D/2D hybrid perovskite can effectively suppress the ion migration due to the morphological advantage observed by scanning electron microscopy (SEM) images (Fig. 3d). As the ion migration mostly occurs on grain boundaries (GBs), mobile ions have to migrate much longer pathway in the granular-like 3D/2D hybrid perovskite than in the columnar-like 3D perovskite (Supplementary Figure 14). Furthermore, the 3D/2D hybrid perovskite has a much greater number of boundary nodes where migrating ions along the direction of the external electric field can be blocked and lose their kinetic energy leading to the termination of their migration. In order to scrutinize the structure of the 3D/2D hybrid perovskites in grain scale, we used high-angle annular dark-field scanning transmission electron microscopy (HAADF-STEM) and clearly observed the 2D perovskite layer with $n = 1$ (Supplementary Figure 15a, b). It is worth noting that the smaller d -spacing (1.15 nm) than that obtained by grazing-incident X-ray diffraction (GIXD) analysis (1.66 nm) can be attributed to transient lattice contraction due to the highly strong beam intensity.^{48,49} Interestingly, the highly periodic 2D perovskite was only observable at the shell region of the grain. Because the 2D perovskite has lower surface energy than the 3D perovskite due to its fewer surface dangling bonds and surface relaxation, growth of the 2D perovskite most possibly occurs on the 3D perovskite grains and it also makes an interface with grain boundary which has the highest surface energy, thereby lowering the total potential energy of the system.^{48,50} Therefore, the mounted 2D perovskite on the 3D perovskite can effectively passivate the traps and block ion migration. We also monitored the composition of the perovskite grains using energy dispersive X-ray spectrometry in scanning TEM mode (EDXS-STEM). HAADF-STEM images and corresponding EDXS elemental maps indicate that the grain is composed of Pb and Br elements (Supplementary Figure 15c-f).

In the Methods,

Transmission electron microscopy (TEM) analysis. High-angle annular dark-field scanning transmission electron microscopy (HAADF-STEM) and energy-dispersive X-ray spectrum (EDXS) imaging were performed on an aberration-corrected Titan-Themis 60-300 (Thermo Scientific) at an acceleration voltage of 200 kV with probe currents of approximately 100 pA and 500 pA, respectively. This microscope is equipped with a high brightness Schottky X-

FEG gun, a Super-X EDX system comprising four silicon drift detectors, and Velox acquisition software. The specimen for STEM analysis was prepared on a carbon support grid by dripping solution which was a dispersion of perovskite powder in chloroform after being scraped from the film.

In the Reference,

48. Ferguson, K. R. *et al.* Transient lattice contraction in the solid-to-plasma transition. *Science Advances* **2**, e1500837 (2016).
49. Maehlen, J. P., Mongstad, T. T., You, C. C. & Karazhanov, S. Lattice contraction in photochromic yttrium hydride. *Journal of Alloys and Compounds* **580**, S119–S121 (2013).
50. Yang, Y., Gao, F., Gao, S. & Wei, S.-H. Origin of the stability of two-dimensional perovskites: a first-principles study. *Journal of Materials Chemistry A* **6**, 14949–14955 (2018).

3. Can the authors prepare 3D/2D structure with protonated BnA and MA?

Answer)

We appreciate the reviewer's valuable comment. In order to see if the use of pre-protonated BnA can lead to the formation of 3D/2D perovskite, we incorporated benzylammonium bromide (BnABr) into MAPbBr₃ precursor with various concentrations. As it is originally charged in the form of a salt and already protonated unlike neutral amines such as aniline (ANI) and benzylamine (BnA), it can be expected that methylammonium (MA⁺) does not transfer its proton to the pre-protonated BnA. We performed ¹H NMR spectroscopy to confirm the absence of proton transfer (Fig. R4a). The spectrum of pristine MAPbBr₃ precursor showed two dominant proton signals, one at $\delta = 7.44$ ppm that represents the ammonium (-NH₃⁺) of MA⁺, and the other at 2.32 ppm that represents its methyl (-CH₃) group. Indeed, the addition of BnABr 2.4 mol% to the MAPbBr₃ precursor did not present evidence of the proton transfer from MA⁺. Even high concentrations of BnABr did not cause a noticeable chemical shift of the proton signal of MA⁺ but only led to the increased intensity of the rest of the signals which can be attributed to the increased amount of the added BnABr. This result indicates that MA⁺ and the added BnABr are present in the solution without causing a chemical reaction such as the proton transfer.

Nevertheless, the incorporation of BnABr into the MAPbBr₃ resulted in the formation of quasi-2D perovskite because BnA is already protonated and ready for being crystallized in

the precursor state. Optical absorbance and photoluminescence (PL) of the perovskite films with different amounts of BnABr were investigated (Fig. R4b-d). The excitonic peak of the MAPbBr₃ absorption spectrum at 525 nm was significantly weakened and blue-shifted from when 30 mol% of BnABr was introduced, which implies that the 3D perovskite cannot maintain its crystalline structure. Instead, the addition of BnABr resulted in the appearance of new excitonic peaks which can be ascribed to the formation of quasi-2D perovskite with a mixed phase.^{R13-R16} Especially, the addition of 100 mol% of BnABr led to the appearance of the excitonic peaks at 399 nm, 430 nm, and 454 nm which can be assigned to a phase with $n = 1, 2,$ and $3,$ respectively. PL result also presented evidence of the quasi-2D perovskite formation by the BnABr addition. Although the addition of 2.4 mol% BnABr did not cause a shift in the peak position compared to the PL of 3D MAPbBr₃ at 543 nm, the addition of the larger amount of BnABr resulted in a significant blue-shift of PL emission (to 540 nm for 30 mol% and to 511 nm for 50 mol%). The further addition of BnABr up to 100 mol% caused the formation of quasi-2D perovskite with a mixed phase. It is worth noting that the addition of BnABr showed the spectrum of quasi-2D perovskite with a mixed phase both in absorbance and PL because MA⁺ which does not deprotonate can still participate in the crystallization of perovskite. In contrast, the use of BnA 100 mol% resulted in those spectra of much purer 2D perovskite due to its strong proton-withdrawing tendency (Fig. 2f and Fig. 3c,d in the revised manuscript).

Fig. R4 ¹H NMR spectra according to the different amounts of **a** BnABr addition into MAPbBr₃ solution precursor solutions in DMSO-*d*₆. **b** Absorption spectra, **c** PL spectra and **d** Normalized PL spectra of MAPbBr₃ film with different amounts of BnABr.

- R13. Xiao, Z. *et al.* Efficient perovskite light-emitting diodes featuring nanometre-sized crystallites. *Nature Photonics* **11**, 108–115 (2017).
- R14. Byun, J. *et al.* Efficient Visible Quasi-2D Perovskite Light-Emitting Diodes. *Advanced Materials* **28**, 7515–7520 (2016).
- R15. Yang, X. *et al.* Efficient green light-emitting diodes based on quasi-two-dimensional composition and phase engineered perovskite with surface passivation. *Nature Communications* **9**, 570 (2018).
- R16. Yang, X. *et al.* Effects of Organic Cations on the Structure and Performance of Quasi-Two-Dimensional Perovskite-Based Light-Emitting Diodes. *J. Phys. Chem. Lett.* **10**, 2892–2897 (2019).

Reviewer #2 (Remarks to the Author):

The authors report on the improvement of characteristics (increase in luminous efficiency, reduced overshoot) and operation lifetime of the hybrid 3D/2D perovskite light-emitting diodes. The improvements were achieved via the introduction of neutral benzylamine (BnA) to methylammonium lead bromide perovskites, and as the author claim, it helped to reduce the formation of deep-traps and ion-migration due to trap-state passivation of 3D perovskite with the 2D perovskite. I have to emphasize that the study is interesting and presented in a well-organized and logical way. In this study, the authors used many experimental techniques that were supplemented with simulation results; this allowed for detailed exploration chemistry of the hybrid 3D/2D perovskite formation, ion migration, charge carrier transport and other essential events in both films and 3D/2D hybrid perovskite LEDs. Given the high quality of the work, I recommend publication of this paper in Nature Communications after addressing the following comments:

1. In addition to benzylamine, the authors also used aniline (ANI) for the formation of the perovskite. But the reasons behind which this molecule was used are not entirely clear; thus, I suggest to provide more details on why ANI was used, this also will give more clarity for the reader.

Answer)

The reason why we used aniline (ANI) is to clearly address the importance of proton-transfer from methylammonium (MA^+) to neutral amine. ANI can be the best example to be compared with BnA because ANI has the same molecular structure except for a methyl bridge between the benzene ring and amine which only BnA has. Interestingly, only such this simple difference resulted in dramatic changes in all the characterizations presented in the manuscript depending on the feasibility of accepting a proton from MA^+ . We additionally described the reason why we used ANI in the manuscript as below according to the reviewer's suggestion.

Revised parts in the manuscript)

In the page 3, line 24,

To compare with BnA, we used aniline (ANI) as it has the same molecular structure except for a methylene bridge between benzene ring and amine. The addition of ANI does not make the structural change maintaining the 3D phase of perovskite as the case without an additive.

2. Figure 3 e shows that the structure of 3D/2D perovskite containing BnA is more grained compared to pristine MAPbBr₃ and MAPbBr₃ with 2.4 mol% of ANI. The authors claim that the presence of small grains can be beneficial for the efficient exciton formation within a grain and provides relevant references. Authors should note that based on more recent studies (10.1021/acsnano.6b02734) exciton binding energies of MAPbBr₃ were found to be similar or even below thermal energies at room temperature, and it is likely that free charges are formed instead of excitons. Overall, the benefit of obtaining grained morphology seems to me not very desirable.

Answer)

We appreciate the reviewer's incisive comment. As the reviewer mentioned, the recent study (10.1021/acsnano.6b02734) reported exciton binding energy (E_b) of MAPbBr₃ as ~ 15.33 meV.^{R1} However, the value was a result of investigating bulk MAPbBr₃ single crystals and the authors described in the article as "*It is worth noting that the investigation of the localized three-dimensional Wannier–Mott excitons in the present study serves as a model that can be rescaled for the study of excitons in low dimensional perovskite materials (nanocrystals, platelets or thin films)*". Thus, E_b of thin film MAPbBr₃ that we developed would be quite different from the reported value. For example, Banerji et al. estimated the E_b of polycrystalline thin film MAPbBr₃ to 110 meV and claimed the co-existence of free carriers and excitons.^{R2} The authors also investigated a polymer- or a small molecule-MAPbBr₃ blend and claimed that the smaller MAPbBr₃ crystals can have a larger E_b that favours the formation of excitons upon photoexcitation because the addition of polymer or small molecule increases its volumetric fraction in the perovskite leading to the changes in the dielectric environment.^{R2} In this regard, the incorporation of TPBi for the additive-based nanocrystal-pinning into MAPbBr₃ in our previous work can contribute to the further increase in the E_b of MAPbBr₃ and the reduced grain size of the 3D/2D hybrid perovskite can enhance the excitonic character. Furthermore, the formation of 2D perovskite in the 3D/2D hybrid perovskite can also additionally increase the E_b .^{R3,R4}

In order to verify which species in our 3D perovskite and 3D/2D hybrid perovskites contributes to emission, we investigated the carrier confinement effect in the perovskites by

performing the excitation density dependent PL measurement at room temperature (Fig. R1). In the excitation density range of 10^{15} - 10^{18} cm^{-3} which corresponds to the operation regime of LEDs,^{R5} The integrate PL (PL_{int}) follows a power-law dependence on the excitation density, n_{ex} as $\text{PL}_{\text{int}} \propto n_{\text{ex}}^k$ where the 3D perovskite had $k = 1.30$ indicating the co-existence of excitons (monomolecular emission) and free carriers (bimolecular emission) while the 3D/2D hybrid perovskite film showed $k = 1.08$ indicating the predominant excitonic character by monomolecular recombination, which can be attributed to the effective carrier confinement.

Fig. R1 Integrated PL as a function of the excitation density of 3D and 3D/2D hybrid perovskites.

- R1. Tilchin, J. *et al.* Hydrogen-like Wannier–Mott Excitons in Single Crystal of Methylammonium Lead Bromide Perovskite. *ACS Nano* **10**, 6363–6371 (2016).
- R2. Droseros, N. *et al.* Origin of the Enhanced Photoluminescence Quantum Yield in MAPbBr₃ Perovskite with Reduced Crystal Size. *ACS Energy Lett.* **3**, 1458–1466 (2018).
- R3. Byun, J. *et al.* Efficient Visible Quasi-2D Perovskite Light-Emitting Diodes. *Advanced Materials* **28**, 7515–7520 (2016).
- R4. Yuan, M. *et al.* Perovskite energy funnels for efficient light-emitting diodes. *Nature Nanotech* **11**, 872–877 (2016).
- R5. Chen, Z. *et al.* Recombination Dynamics Study on Nanostructured Perovskite Light-Emitting Devices. *Adv. Mater.* **30**, 1801370 (2018).

Revised parts in the manuscript)

In the page 9, line 22,

We further investigated the carrier confinement effect in the perovskites by performing the excitation density dependent PL measurement at room temperature (Supplementary Figure 9).

In the excitation density range of 10^{15} - 10^{18} cm^{-3} which corresponds to the operation regime of LEDs,³³ The integrate PL (PL_{int}) follows a power-law dependence on the excitation density, n_{ex} as $\text{PL}_{\text{int}} \propto n_{\text{ex}}^k$ where the 3D perovskite had $k = 1.30$ indicating the co-existence of excitons (monomolecular emission) and free carriers (bimolecular emission) while the 3D/2D hybrid perovskite film showed $k = 1.08$ indicating the predominant excitonic character by monomolecular recombination, which can be attributed to the effective carrier confinement.³⁴

In the Reference,

33. Chen, Z. *et al.* Recombination Dynamics Study on Nanostructured Perovskite Light-Emitting Devices. *Adv. Mater.* **30**, 1801370 (2018).
34. Droseros, N. *et al.* Origin of the Enhanced Photoluminescence Quantum Yield in MAPbBr₃ Perovskite with Reduced Crystal Size. *ACS Energy Lett.* **3**, 1458–1466 (2018).

In the Supplementary Figure 9,

Supplementary Figure 9 Integrated PL as a function of the excitation density of 3D and 3D/2D hybrid perovskites.

3. Figure 4 shows that the PL intensity changes depending on the perovskite composition and additive concentrations. It is evident that the PL intensity is very sensitive on the position of the sample which makes it difficult to measure its intensity accurately; thus for the accurate estimation of the intensity differences, PL Quantum yield measurements would be more useful.

Answer)

We appreciate the reviewer's valuable comment. To confirm the steady-state photoluminescence (PL) result, we additionally performed the absolute PL quantum efficiency (PLQE) measurement (Fig. R2). The PLQE result showed a similar trend to the steady-state PL result according to the amount of BnA. When the small amount of BnA (2.4 mol%) was added into MAPbBr₃, the film showed the highest PL intensity and PLQE, and a further increase of BnA up to 30 mol% led to a decrease in the PL intensity and the PLQE. However, when a large amount of BnA (50 mol%) was added, the PL intensity and the PLQE of the film increased again. We may attribute this behaviour to a phase transition of perovskite from 3D/2D hybrid phase to quasi-2D phase as the amount of BnA increases. To be specific, the increase in the amount of BnA in the 3D/2D hybrid phase regime did not cause a significant shift of PL peak position because the 3D perovskite can maintain its crystalline structure. However, the increasing amount of 2D perovskite can relatively decrease the amount of 3D perovskite reducing the PL intensity of 3D perovskite and PLQE. In the quasi-2D regime, on the other hand, an increase in the amount of BnA caused a significant blue shift of PL peak position to 536 nm for 30 mol% and 530 nm for 50 mol%. The recovered PL intensity and the PLQE in the quasi-2D regime at 50 mol% can be attributed to the optimized energy funnelling and confinement effect in the quasi-2D perovskite structure.^{R6} The relatively lower PLQE compared to our previous report can be attributed to the use of low-power laser diode (1.98 mW) due to the limitation of access to the previous set-up.^{R7} We added a brief description of the phase transition and included the PLQE result in the manuscript to support and confirm the steady-state PL result.

Fig. R2 (a) PL spectra, (b) PL peak position and (c) PLQE of MAPbBr₃ film with different amounts of BnA.

- R6. Yang, X. *et al.* Efficient green light-emitting diodes based on quasi-two-dimensional composition and phase engineered perovskite with surface passivation. *Nature Communications* **9**, 570 (2018).
- R7. Cho, H. *et al.* Overcoming the electroluminescence efficiency limitations of perovskite light-emitting diodes. *Science* **350**, 1222–1225 (2015).

Revised parts in the manuscript)

In the page 9, line 3,

Only the addition of the larger amount of BnA caused a blue-shift of PL emission (to 537 nm for 30 mol% and to 530 nm for 50 mol%), accompanied by the appearance of 2D perovskite PL emissions at 405 nm and 435 nm, which can be assigned to $n = 1$ and 2, respectively. The further addition of BnA up to 100 mol% which corresponds to the same amount as MA, caused a dramatic increase in the intensity of PL emission at 409 nm from 2D perovskite. This shift can be more clearly seen in normalized PL spectra (Fig. 4d). **We also measured the absolute PL quantum efficiency (PLQE) which showed a similar trend to the steady-state PL intensity according to the amount of BnA (Supplementary Figure 5).** In contrast, however, increasing the amount of ANI only decreased the PL intensity, and did not shift the emission wavelength, which can be attributed to intensive non-radiative recombination due to the electron deficiency of ANI (Supplementary Figure 6).³⁰

In the Supplementary Figure 5,

Supplementary Figure 5 PLQE of MAPbBr₃ film according to the amount of BnA addition. The films were excited with a continuous-wave 405 nm laser diode (2.5 mW cm⁻²).

In the Methods,

PLQE of perovskite thin films on a quartz substrate was measured by using the same integrating sphere method reported previously with a continuous-wave 405 nm laser diode (2.5 mW cm^{-2}).

4. The hole-only devices employed molybdenum oxide layer on top of the perovskite. However, information on how these devices were made and MoO₃ layer was deposited is missing.

Answer)

We appreciate the reviewer's valuable comment. As the reviewer mentioned, we added the relevant method of hole-only device fabrication.

Revised parts in the manuscript)

In the Methods,

Device fabrication. For PeLEDs, Buf-HIL layer was spin-coated on a cleaned FTO-coated glass as previously reported.³ On the prepared substrate, the prepared perovskite precursor solution was spin-coated at 3000 rpm, followed by the additive-based nanocrystal-pinning process to induce immediate crystallization of perovskite by using TPBi solution in chloroform.³ The deposited perovskite film was annealed on a hot plate at 90 °C for 10 min. Subsequently, 50 nm of TPBi, 1 nm of LiF, and 100 nm of Al were thermally evaporated in sequence in a high-vacuum chamber to complete the device. For HODs, 30 nm of molybdenum oxide (MoO₃) and 50 nm of Au were thermally deposited on top of MAPbBr₃ to make a structure of glass/FTO/Buf-HIL/MAPbBr₃/MoO₃/Au in the high-vacuum chamber. The devices were encapsulated in N₂ atmosphere and the pixel area of a device was 4.8 mm².

Minor issues:

- 1. Please, define what is Buf-HIL when first time used in the text.**
- 2. Line 201. Unknown character after the "type-".**
- 3. Some references (5; 14; 26; 46;) should be corrected since complete bibliographic information is missing.**

Answer)

We appreciate the reviewer's valuable comment. We correctly revised the corresponding issues in the manuscript as below.

Revised parts in the manuscript)

In the page 8, line 1,

The perovskite films were prepared on a glass substrate/**buffer hole-injection layer (Buf-HIL)**, which consists of **poly(3,4-ethylenedioxythiophene) polystyrene sulfonate (PEDOT:PSS)** and perfluorinated ionomer.

In the page 9, line 5,

Ultraviolet photoemission spectroscopy (UPS) analysis of each 3D MAPbBr₃ and 2D BnA₂PbBr₄ film confirmed the **type-I** quantum well band alignment of 3D/2D hybrid perovskite (Supplementary Figure 7 and Supplementary Table 4).

In the Reference,

5. Zhao, B. *et al.* High-efficiency perovskite–polymer bulk heterostructure light-emitting diodes. *Nature Photon* **12**, 783–789 (2018).
19. Shang, Y., Li, G., Liu, W. & Ning, Z. Quasi-2D Inorganic CsPbBr₃ Perovskite for Efficient and Stable Light-Emitting Diodes. *Adv. Funct. Mater.* **28**, 1801193 (2018).
25. Lee, H.-D. *et al.* Efficient Ruddlesden–Popper Perovskite Light-Emitting Diodes with Randomly Oriented Nanocrystals. *Advanced Functional Materials* **29**, 1901225 (2019).
54. Zhou, R. Carbon Nanotubes. in *Modeling of Nanotoxicity* 45–59 (Springer International Publishing, 2015).

Reviewer #3 (Remarks to the Author):

The authors report on the role of 3D/2D hybrid perovskite materials in suppressing ion migration, which, as a result, reduces the overshoot of luminance over time as compared to 3D MAPbBr₃ perovskites in perovskite LEDs. This was achieved by adding neutral benzylamine that was assumed to induce proton transfer from MA to facilitate the crystallization of the 2D perovskite, which is, as a result, claimed to suppress the formation of trap states and ion migration, enhancing operational stability and efficiency. While the study presented is very insightful in general and impactful for the community, a number of claims have not been rigorously assessed and remain speculative, such as the role of proton transfer in the entire process, although highlighted throughout the manuscript. For this purpose, this reviewer recommends a major revision of the study before considering further for publication. Some of the critical remarks that should be addressed are the following.

1. The authors highlight the role of proton-transfer in the analyzed properties without providing direct evidence that such a process is instrumental in the solid-state. Specifically, although solution-based NMR spectra provide valuable insights, this does not provide evidence in the solid-state. Moreover, it remains unclear whether the proton transfer is a critical factor determining the properties or the usage of benzylamine leads to the observed effects. For this purpose, molecular dynamics simulations in conjunction with control experiments employing benzylammonium salts would be helpful. Importantly, the effects should be described without excluding other contributing factors in the analysis.

Answer)

We appreciate the reviewer's valuable comment. As the reviewer mentioned, solution-state NMR supports the evidence of the proton transfer from methylammonium (MA⁺) to benzylamine (BnA) in the precursor state of the perovskite prior to its crystallization. However, once the crystallization occurs, in other words, in the solid-state of the perovskite, we cannot directly examine the proton-transfer anymore because the perovskite already forms its solid-state lattice composed of corresponding ions mainly with strong ionic bonds. Thus, we investigated the formation of 2D perovskite in the solid-state by various suitable tools such as UV-Vis absorption, photoluminescence (PL), X-ray diffraction

(XRD), and scanning electron microscopy (SEM), which presented subsequent results of the proton-transfer well.

As a matter of fact, we introduced aniline (ANI) for comparison to verify the importance of the proton transfer because it does not lead to the proton transfer. The reason for this unfeasible proton transfer from MA^+ to ANI was explained in the manuscript with electron deficiency of ANI due to a resonance effect that originates from its molecular structure. As we presented, no evidence of 2D perovskite formation was found by using ANI indicating that the protonation is necessary to result in the 2D perovskite formation.

First-principles molecular dynamics (MD) simulations also presented the importance of proton-transfer. We compared thermal fluctuation behaviours of perovskite lattice including protonated or unprotonated BnA at 300K (Fig. R1). The atoms in the 2D perovskite formed by the protonated BnA (BnA^+) only slightly vibrate around their equilibrium positions and show steady energy fluctuations, which suggests that the 2D structure is well maintained. On the contrary, the 2D perovskite formed by the unprotonated BnA showed highly significant oscillations of the fluctuations of total energy, which indicates the intrinsic instability of the system. This result supports the importance of the protonation which enables BnA^+ to have a strong bond and implies that the incorporation of the protonated BnA (BnA^+) can lead to effective passivation because it can maintain the 3D/2D hybrid structure without forming defects in the lattice.

Fig. R1 Fluctuations of total energy as the evolution of simulation time and the snapshots of atomic configurations after the first-principles molecular dynamics (MD) simulations (3.5 ps) with a time step of 1.2 fs at the temperature of 300 K. **a** and **b** for the 2D perovskites formed by protonated BnA (BnA^+) and unprotonated BnA, respectively.

Fig. R2 ¹H NMR spectra according to the different amounts of **a** BnABr addition into MAPbBr₃ solution precursor solutions in DMSO-*d*₆. **b** Absorption spectra, **c** PL spectra and **d** Normalized PL spectra of MAPbBr₃ film with different amounts of BnABr.

According to the reviewer's suggestion, we performed additional experiments by using benzylammonium bromide (BnABr). As it is originally charged in the form of a salt and already protonated unlike neutral amines such as aniline (ANI) and benzylamine (BnA), it can be expected that methylammonium (MA⁺) does not transfer its proton to the pre-protonated BnA. We performed ¹H NMR spectroscopy to confirm the absence of proton transfer (Fig. R2a). The spectrum of pristine MAPbBr₃ precursor showed two dominant proton signals, one at δ = 7.44 ppm that represents the ammonium (-NH₃⁺) of MA⁺, and the other at 2.32 ppm that represents its methyl (-CH₃) group. Indeed, the addition of BnABr 2.4 mol% to the MAPbBr₃ precursor did not present evidence of the proton transfer from MA⁺. Even high concentrations of BnABr did not cause a noticeable chemical shift of the proton signal of MA⁺ but only led to the increased intensity of the rest of the signals which can be attributed to the increased amount of the added BnABr. This result indicates that MA⁺ and the added BnABr are present in the solution without causing a chemical reaction such as the proton transfer.

To see if the addition of BnABr can lead to the 2D perovskite formation, we fabricated films using the corresponding precursor solutions and investigated optical absorbance and photoluminescence (PL) of the perovskite films with different amounts of BnABr (Fig. R2b-d). The incorporation of BnABr into the MAPbBr₃ resulted in the formation of quasi-2D perovskite because BnA is already protonated and ready for being crystallized in the precursor state. The excitonic peak of the MAPbBr₃ absorption spectrum at 525 nm was significantly weakened and blue-shifted from when 30 mol% of BnABr was introduced, which implies that the 3D perovskite cannot maintain its crystalline structure. Instead, the addition of BnABr resulted in the appearance of new excitonic peaks which can be ascribed to the formation of quasi-2D perovskite with a mixed phase.^{R1-R4} Especially, the addition of 100 mol% of BnABr led to the appearance of the excitonic peaks at 399 nm, 430 nm, and 454 nm which can be assigned to a phase with $n = 1, 2,$ and $3,$ respectively. PL result also presented evidence of the quasi-2D perovskite formation by the BnABr addition. Although the addition of 2.4 mol% BnABr did not cause a shift in the peak position compared to the PL of 3D MAPbBr₃ at 543 nm, the addition of the larger amount of BnABr resulted in a significant blue-shift of PL emission (to 540 nm for 30 mol% and to 511 nm for 50 mol%). The further addition of BnABr up to 100 mol% caused the formation of quasi-2D perovskite with a mixed phase. It is worth noting that the addition of BnABr showed the spectrum of quasi-2D perovskite with a mixed phase both in absorbance and PL because MA⁺ which does not deprotonate can still participate in the crystallization of perovskite. In contrast, the use of BnA 100 mol% resulted in those spectra of much purer 2D perovskite due to its strong proton-withdrawing tendency (Fig. 2f and Fig. 3c,d in the revised manuscript).

To underline the importance of the protonation, we included the MD simulation result in the manuscript and Supplementary information as below.

- R1. Xiao, Z. *et al.* Efficient perovskite light-emitting diodes featuring nanometre-sized crystallites. *Nature Photonics* **11**, 108–115 (2017).
- R2. Byun, J. *et al.* Efficient Visible Quasi-2D Perovskite Light-Emitting Diodes. *Advanced Materials* **28**, 7515–7520 (2016).
- R3. Yang, X. *et al.* Efficient green light-emitting diodes based on quasi-two-dimensional composition and phase engineered perovskite with surface passivation. *Nature Communications* **9**, 570 (2018).

R4. Yang, X. *et al.* Effects of Organic Cations on the Structure and Performance of Quasi-Two-Dimensional Perovskite-Based Light-Emitting Diodes. *J. Phys. Chem. Lett.* **10**, 2892–2897 (2019).

Revised parts in the manuscript)

In the page 6, line 20,

Furthermore, we predicted the importance of the protonation of BnA by performing first-principles molecular dynamics (MD) simulations in which perovskite lattice including the protonated BnA (BnA^+) showed much greater structural stability than that with neutral BnA. (Supplementary Figure 2).

In the Supplementary Figure 2,

Supplementary Figure 2 Fluctuations of total energy as the evolution of simulation time and the snapshots of atomic configurations after the first-principles molecular dynamics (MD) simulations (3.5 ps) with a time step of 1.2 fs at the temperature of 300 K. **a** and **b** for the 2D perovskites formed by protonated BnA (BnA^+) and unprotonated BnA, respectively.

First-principles molecular dynamics (MD) simulations presented the importance of proton-transfer. We utilized a $2\sqrt{2} \times 2\sqrt{2} \times 1$ supercell to compare thermal fluctuation behaviors of perovskite lattice including protonated or unprotonated BnA at 300K. The atoms in the 2D perovskite formed by the protonated BnA (BnA^+) only slightly vibrate around their equilibrium positions and show steady energy fluctuations, which suggests that the 2D structure is well maintained. On the contrary, the 2D perovskite formed by the unprotonated BnA showed highly significant oscillations of the fluctuations of total energy, which indicates the intrinsic instability of the system. This result supports the importance of the

protonation which enables BnA^+ to form a strong bond in the lattice and implies that the incorporation of the protonated BnA (BnA^+) can lead to effective passivation since it can maintain the 3D/2D hybrid structure without forming defects in the lattice.

2. The authors should clarify in the main manuscript whether monitoring the protonation reaction via UV-vis spectroscopy was performed in solution or in the corresponding thin-films.

Answer)

We appreciate the reviewer's valuable comment. The UV-vis spectroscopy was performed with thin films to see the resulting 2D perovskite formation, which we already described in the manuscript as "*The absorption characteristics of the perovskite films from the corresponding precursors support this result (Fig. 2f, g): the excitonic peak of the MAPbBr_3 absorption spectrum at 525 nm changes significantly according to the amount of BnA added, but does not shift with the addition of ANI regardless of the amount.*" and in the Methods section as "***Characterizations of perovskite films.*** The absorbance of the perovskite film was measured using UV-vis absorption spectrophotometer (Cary-5000)".

3. The study concludes that the higher propensity of the benzylamine to form the 2D perovskite phase as compared to the aniline is based on the difference in their corresponding pKa. This is, however, rather misleading, as the geometry of the molecules also defines the ability to effectively form layered 2D phases. In this regard, the presence of the methylene linker enables effective and adaptable penetration into the A-cation vacancy for the benzylammonium system, unlike the anilinium, which would result in the formation of a better-defined layered structure. In fact, anilinium-based 2D phase could co-exist, yet its poor orientation/crystallinity would prevent it from being observed experimentally via X-ray diffraction. Considering these factors, the authors are strongly advised not to exclusively ascribe the observed properties to the proton transfer effects. This is particularly misleading in the title of the manuscript.

Answer)

We appreciate the reviewer's incisive comment. We agree that the geometric difference of additives can influence the crystallization of perovskite as the reviewer

explained. However, one can take the geometric factor into account only when an additive is protonated in the precursor state and then it can actually participate in forming a chemical bond in the solid-state perovskite lattice. However, ANI cannot be protonated in MAPbBr₃ precursor solution due to its electron deficiency based on a resonance effect that originates from its molecular structure as we described in the manuscript. Therefore, ANI cannot take part in the crystallization or making a chemical bond in the solid-state perovskite film.

To support our claim, we compared the optical characteristics of perovskite films using ANI and ANIBr. We investigated the absorption and the PL characteristics of the MAPbBr₃ perovskite films with ANI or ANIBr. In contrast to ANI, the use of ANIBr can yield protonated ANI (ANI⁺) in the precursor state as it is originally charged in the form of a salt. Thus, we can exclusively examine the effect of protonation (Fig. R3). As we already presented in the manuscript, the addition of ANI did not cause a significant shift of the excitonic absorption peak or PL emission wavelength of the MAPbBr₃ film regardless of its amount, but only decreased the intensity of PL. This result indicates that ANI does not participate in a chemical reaction or make a chemical bond, but exist as an impurity. In contrast, the use of ANIBr showed a clear change both in the absorption and the PL characteristics. The excitonic absorption peak and the PL peak of MAPbBr₃ shifted towards the shorter wavelength by increasing the amount of ANIBr. The diminished excitonic absorption peak implies that the incorporation of ANIBr influenced the crystallization of perovskite degrading its crystallinity. Although there was no evidence of the formation of low-dimensional perovskites from the absorption spectra, the addition of ANIBr led to an improvement in PL intensity accompanied by the blue-shift of the emission spectrum. The enhanced PL intensity can be ascribed to defect passivation as the protonated aniline (ANI⁺) can interact with the surface terminal [PbBr₆]⁴⁻ octahedra via hydrogen bonding,^{R5,R6} and the blue shift in the PL spectra might be caused by the reduced size of crystallites.^{R7} We may attribute the absence of features of the 2D perovskites upon adding ANIBr unlike BnABr to its different molecular geometry due to the methylene linker. Further study regarding how the molecular geometric difference of additives molecules influence on perovskite materials using protonated organic molecules such as ANIBr and BnABr would be greatly important.

Fig. R3 UV-Vis absorption, PL, and normalized PL spectra of MAPbBr₃ film with ANI (a-c) and ANIBr (d-f) with different amounts of each additive.

- R5. Li, X. *et al.* Improved performance and stability of perovskite solar cells by crystal crosslinking with alkylphosphonic acid ω -ammonium chlorides. *Nature Chemistry* **7**, 703–711 (2015).
- R6. Ming Koh, T. *et al.* Enhancing moisture tolerance in efficient hybrid 3D/2D perovskite photovoltaics. *Journal of Materials Chemistry A* **6**, 2122–2128 (2018).
- R7. Xiao, Z. *et al.* Efficient perovskite light-emitting diodes featuring nanometre-sized crystallites. *Nature Photonics* **11**, 108–115 (2017).

4. The reduced reorientation rate of the benzylammonium system is ascribed to the π - π interactions of the neighboring rings. In addition, however, the authors should also consider the steric repulsion of the molecules that should be mentioned.

Answer)

We appreciate the reviewer’s perceptive comment. We added a corresponding description in terms of the effect of steric repulsion on the molecular reorientation in the manuscript as below.

Revised parts in the manuscript)

In the page 16, line 19,

In this regard, BnA⁺ can have remarkably reduced reorientation rate by themselves based on their π - π interaction and steric repulsion as a result of neighbouring benzene rings; this slowing of BnA⁺ reorientation in the perovskite lattice can effectively impede ions migration along GBs, compared to MA⁺, which has a high rotational degree of freedom.

5. The analysis of the effect of the additives on the optoelectronic properties focuses on the comparison of the benzylamine-containing system and the pristine perovskite. However, the authors should also perform a control study involving aniline-containing perovskite materials, as, despite the lack of apparent effect on the structural properties, their presence could suppress ion migration and affect the performance.

Answer)

We appreciate the reviewer's constructive comment. According to the reviewer's suggestion, we additionally conducted control experiments using aniline (ANI). We used the same device structure except for the perovskite emitter in which 2.4 mol% ANI was added. All LED characteristics were significantly degraded in the device with the emitter compared to those of the device with pristine MAPbBr₃ emitter. The PeLED with ANI-added emitter had maximum current efficiency $CE_{\max} = 6.81 \text{ cd}\cdot\text{A}^{-1}$, and external quantum efficiency $EQE_{\max} = 1.37 \%$, which are lower than those of the PeLED with pristine MAPbBr₃ emitter ($CE_{\max} = 13.23 \text{ cd}\cdot\text{A}^{-1}$ and $EQE_{\max} = 2.50 \%$) (Fig. R4). The degradation of the LED characteristics of the PeLED using the ANI-added emitter can be attributed to significant non-radiative recombination by the electron-deficient ANI as well as by the ionic space charges that are distributed at the vicinity of perovskite interfaces because the addition of ANI cannot contribute to suppression of the ion migration.^{R8}

Fig. R4 EL characteristics of PeLEDs using pristine MAPbBr₃ emitter (red) and 2.4 mol % ANI-added MAPbBr₃ emitter (blue). **a** Current density of PeLEDs as a function of the applied voltage. **b** Luminance of PeLEDs as a function of the applied voltage. **c** External quantum efficiency (EQE) of PeLED as a function of the applied voltage. **d** Current efficiency (CE) of PeLEDs as a function of the applied voltage.

We also investigated the operational stability of the PeLED using ANI-added MAPbBr₃, which overshoot to 146.7 % of the initial luminance, then dropped to 50 % of the luminance after 19 min while the PeLED using pristine MAPbBr₃ had $T_{50} = 32$ min after reaching 150.4 % of the initial luminance (Fig. R5). The similar level of luminance overshoot of the ANI-added PeLED implies that the addition of ANI cannot effectively suppress the ion migration. We included these additional results in Supplementary information.

Fig. R5 Relative lumiance of PeLEDs using pristine MAPbBr₃ emitter and 2.4 mol % ANI-added MAPbBr₃ emitter over time.

R8. He, C. *et al.* Turn on fluorescence sensing of vapor phase electron donating amines via tetraphenylporphyrin or metallophenylporphrin doped polyfluorene. *Chemical Communications* **46**, 7536–7538 (2010).

Revised parts in the manuscript)

In the page 13, line 22,

In contrast, the use of ANI-added perovskite film degraded the LED characteristics due to significant non-radiative recombination by the electron-deficient ANI as well as by the ionic space charges at the interfaces with the perovskite emitting layer (Supplementary Figure 13).

In the Supplementary Figure 13,

Supplementary Figure 13 EL characteristics of PeLEDs using pristine MAPbBr₃ emitter (red) and 2.4 mol % ANI-added MAPbBr₃ emitter (blue). **a** Current density of PeLEDs as a function of the applied voltage. **b** Luminance of PeLEDs as a function of the applied voltage. **c** External quantum efficiency (EQE) of PeLED as a function of the applied voltage. **d** Current efficiency (CE) of PeLEDs as a function of the applied voltage.

In the page 18, line 20,

We also investigated the operational stability of the PeLED using the ANI-added MAPbBr₃, which showed strong overshoot as 146.7 % compared to the initial luminance, implying that the addition of ANI cannot effectively suppress the ion migration (Supplementary Figure 17).

Supplementary Figure 17,

Supplementary Figure 17 Relative luminance of PeLEDs using pristine MAPbBr₃ emitter and 2.4 mol % ANI-added MAPbBr₃ emitter over time.

6. The literature mostly appears appropriate. However, the work of Loi et al. (*Appl. Phys. Rev.* 2019, 6, 031401), for instance, has not been cited, despite reporting on the relevant usage of benzylamine in hybrid perovskites. In addition, the role of adaptability through the introduction of a flexible linker in the formation of layered perovskite structures should be referred to (for instance, general role of the linker: *CrystEngComm*, 2010, 12, 2646 and the recent methylene group case: *Adv. Energy Mater.* 2019, 1900284).

Answer)

We appreciate the reviewer's valuable comment. We accordingly added the relevant references in the manuscript.

Revised parts in the manuscript)

In the page 3, line 15,

In a conventional way to prepare low-dimensional layered perovskites, an ammonium halide salt with a bulky backbone (e.g. phenethylammonium halide (PEAX), or *n*-butylammonium halide (*n*-BAX)) and the salts of a 3D perovskite (e.g. methylammonium halide (MAX) and lead halide (PbX₂)) have been dissolved together in a solvent or a mixed solvent.^{8,14-19}

In the Reference,

17. Cheng, Z. & Lin, J. Layered organic–inorganic hybrid perovskites: structure, optical properties, film preparation, patterning and templating engineering. *CrystEngComm* 12, 2646–2662 (2010).

18. Zhou, N. *et al.* The Spacer Cations Interplay for Efficient and Stable Layered 2D Perovskite Solar Cells. *Advanced Energy Materials* **10**, 1901566 (2020).

In the page 3, line 19,

Here, unlike the conventional way, we only incorporated a small amount of a liquid-form neutral reagent, benzylamine (BnA) into the methylammonium lead bromide (MAPbBr₃) precursor to induce proton transfer from MA⁺ to BnA;²⁰ as a result, a 3D/2D hybrid dimensional crystal structure formed unlike the conventional quasi-2D perovskites (Fig. 1).

In the Reference,

20. Duim, H. *et al.* Mechanism of surface passivation of methylammonium lead tribromide single crystals by benzylamine. *Applied Physics Reviews* **6**, 031401 (2019).

Thank you for considering the aforementioned remarks, which I believe would contribute to enhancing the impact of this interesting study.

Reviewers' comments:

Reviewer #1 (Remarks to the Author):

Authors have addressed the points raised in the previous review with varying degrees of satisfaction. These results should provide an efficient way for enhancing both the structure stability and luminous performances of halide perovskites. So I would be happy to see the manuscript accepted for publication at Nature Communications.

Reviewer #2 (Remarks to the Author):

I would like to emphasize that the authors have addressed the main concerns, and thus, I consider that the manuscript fulfills all requirements to be published in Nature Communications.

Reviewer #3 (Remarks to the Author):

The authors have attempted to revise some of the critical points presented earlier by this reviewer and improved the manuscript. However, some of the points have not been thoroughly addressed, which prompts the reviewer to further request a revision mainly due to some of the interpretations of results that remain misleading.

1) The authors highlight the role of proton transfer in the process without providing direct evidence that it occurs in the solid-state. While solution-based NMR spectra provide some insights, this does not provide evidence of the occurrence in the solid-state, whereas some of the other methods listed (such as UV-vis spectroscopy, for instance) could only provide a qualitative indication of low-dimensional structure formations without atomic-level insight. This must be clearly stated in the manuscript - the protonation is likely one of the underlying processes but necessarily the exclusive underlying reason. This is particularly concerning as the mechanism has been highlighted in the title.

2) Similarly, the authors continue to refer to the inability of the anilinium system to form a low-dimensional structure due to the difference in basicity. However, the geometry of this system might not be suitable for the formation of a well-defined low-dimensional structure and this must be clearly stated. In fact, an anilinium-based 2D phase could co-exist, yet its poor orientation/crystallinity would prevent it from being observed experimentally via X-ray diffraction. Considering these factors, the authors should not exclusively ascribe the observed properties to the proton transfer effects.

3) The reduced reorientation rate of benzylammonium is ascribed to the π - π interactions and steric repulsion of the neighboring rings. Why does the reorientation impede ion migration? This claim is unclear and requires more clarity.

4) The sentence "3D/2D hybrid dimensional crystal structure formed unlike the conventional quasi-2D perovskites" implies differences between the two structures that have not been defined in the manuscript. This must be appropriately addressed, without inducing further confusion.

5) In summary, in accordance with the previous revision, this reviewer is supportive of this study. However, there is no unambiguous evidence that the protonation is the exclusive factor determining the formation of low-dimensional perovskite phases and suppressing the ion migration. Therefore, the "proton-transfer induced 2D/3D hybrid perovskites" term can be misleading, particularly in the title.

The authors should thus refrain from using this general language and rather highlight the use of a specific system in this context. For instance, "Benzylamine-Mediated 2D/3D Hybrid Perovskite Formation Reduces Luminance Overshoot" appears more appropriate, without any need for using the term "extremely" in this context either. Other variants are most welcome as well, this is a very exciting study and this reviewer strives to ensure that it reaches the impact it deserves.

I believe that it is our responsibility as scientists to contribute to a rigorous scientific practice by using more accurate descriptors instead of coining terms with the objective of publicizing work even at the expense of this being scientifically misleading. This work is already impactful without adding unsubstantiated claims – this reviewer urges the authors to address this carefully to allow the work to express its impact.

I appreciate your consideration.

Answers to Reviewers

First of all, we would like to appreciate the reviewer's valuable comments. We answered the reviewer's questions as below. We also revised our manuscript to comply with the reviewer's comments.

Reviewer #3

The authors have attempted to revise some of the critical points presented earlier by this reviewer and improved the manuscript. However, some of the points have not been thoroughly addressed, which prompts the reviewer to further request a revision mainly due to some of the interpretations of results that remain misleading.

1) The authors highlight the role of proton transfer in the process without providing direct evidence that it occurs in the solid-state. While solution-based NMR spectra provide some insights, this does not provide evidence of the occurrence in the solid-state, whereas some of the other methods listed (such as UV-vis spectroscopy, for instance) could only provide a qualitative indication of low-dimensional structure formations without atomic-level insight. This must be clearly stated in the manuscript - the protonation is likely one of the underlying processes but necessarily the exclusive underlying reason. This is particularly concerning as the mechanism has been highlighted in the title.

Answer)

We appreciate the reviewer's valuable comment. As the reviewer mentioned, the result of the liquid-state NMR spectroscopy evidently showed the occurrence of the proton transfer from methylammonium (MA^+) to benzylamine (BnA) in the precursor state of the perovskite. Nevertheless, the reviewer is intrigued to know regarding its occurrence in the solid-state perovskite. In this paper, we do not claim the necessity of the proton transfer in the solid state but that the proton transfer is a preceding process and prerequisite which enable the neutral reagent, BnA, to take part in the crystallization and the formation of 2D perovskites.

However, to address the reviewer's comments more clearly, we additionally conducted solid-state magic-angle-spinning (MAS) NMR spectroscopy. Figure R1 shows the ^1H MAS NMR spectra for solid-state perovskites with varying BnA concentration (0, 2.4, 30,

50 and 100%) and ANI concentration (0, 2.4, and 100%) under a fast spinning speed of 40 kHz. The pristine sample showed two ^1H peaks at ~ 6.3 ppm and ~ 3.3 ppm which can be assigned to the hydrogens in NH_3 (H-NH $_3$) and CH_3 (H-CH $_3$) of MA^+ , respectively.^{R1} With the addition of BnA, the spectrum had two more peaks that arose from the hydrogens bonded to the benzene ring (H-B) and in CH_2 (H-CH $_2$) of BnA^+ . The positions of H-B varied in the range of ~ 6.7 to ~ 8 ppm and the peak of H-CH $_2$ was at ~ 4.8 ppm.^{R2} The ^1H peak intensity of H-CH $_3$ decreased with an increase in BnA concentration because MA^+ is deprotonated to be methylamine which can be easily evaporated during the film annealing process. Meanwhile, the peak intensities of H-B, H-NH $_3$, and H-CH $_2$ gradually increased confirming that the protonated BnA (BnA^+) composes the solid-state crystalline perovskite and that the proton transfer from MA^+ to BnA indeed occurred. Upon the addition of ANI, in contrast, the peak positions of H-NH $_3$ and H-CH $_3$ were invariant indicating ANI did not participate in the formation of perovskite.

Fig. R1 ^1H MAS NMR spectra (at 14.1 T and a spinning speed of 40 kHz) according to the different amounts of (a) BnA and (b) ANI.

It should be noted that a neutral molecule as it is neither a donor nor acceptor, cannot make a bond or participate in the formation of a solid-state ionic crystal such as perovskites. In other words, a basis in a lattice which is generally an atom or a molecule should be

protonated as such examples of Cs^+ , MA^+ , formamidinium (FA^+), and phenylethylammonium (PEA^+) in order to occupy the so-called 'A-site' in a perovskite lattice. In our case, MA^+ and the protonated BnA (BnA^+) respectively compose the 3D perovskite and the 2D perovskite lattice, and both maintain the solid-state lattice structure based on the strong ionic bonds.

All the results in our work showed great consistency to present that BnA can be protonated but ANI cannot. Additionally, we compared the use of ANI and ANIBr in MAPbBr_3 and investigated the absorption and PL characteristics of the films to solely verify the effect and importance of the protonation. In contrast to ANI, ANIBr can yield protonated ANI (ANI^+), anilinium, in the precursor state as it is originally charged in the form of a salt. Thus, we can exclusively examine the effect of protonation (Fig. R2a-f). Indeed, the use of ANIBr led to a clear blue shift of the excitonic absorption peak and PL spectra, which implies a structural change of perovskites despite the absence of peaks indicating well-defined low-dimensional perovskites. Moreover, the significant improvement in PL intensity with the use of 2.4 mol% of ANIBr can be ascribed to defect passivation as the protonated aniline (ANI^+), anilinium, can interact with $[\text{PbBr}_6]^{4-}$ octahedra via hydrogen bonding.^{R3,R4} In contrast, the addition of the neutral ANI did not present any noticeable shift in the spectra even with 100 mol%, but only led to PL quenching. This indicates that ANI does not involve a chemical reaction or a chemical bond, but does exist as an impurity or a PL quencher. If we assume that the neutral ANI can be protonated, there would and should have been a perceptible change in the spectra as shown with the use of ANIBr.

The influence of the molecular geometry that the reviewer addressed in the second comment did not have to be considered in this comparison where ANI remains neutral not being able to participate in the crystallization of perovskite because the proton-transfer process was the main factor that precedes the crystallization of perovskite. The geometric factor of the molecules which influences a resulting perovskite structure can be considered only when the molecule is able to be crystallized by interacting with the rest of the species such as Pb^{2+} and Br^- . As ANI cannot be protonated in MAPbBr_3 precursor solution, however, the argument concerning the molecular geometry with ANI would not be necessary.

Instead, a comparison between the use of ANIBr and BnABr can give the insight about the influence of the molecular geometry because both ammoniums are pre-protonated in the form of a salt but they have different molecular structure. Unlike the use of ANIBr which was not sufficiently suitable for the formation of a well-defined low-dimensional phase, the incorporation of BnABr presented a typical feature of quasi-2D perovskite resulting in the appearance of new excitonic peaks in absorption and PL spectra accompanied by blue shift

(Fig. R2g-i).^{R5-R8} This result shows how the molecular geometry impacts on the formation of perovskites.

Fig. R2 UV-Vis absorption, PL, and normalized PL spectra of MAPbBr₃ films with ANI (a-c), ANIBr (d-f), and BnABr (g-i) with different amounts of addition.

Table R1 Summary of a overall comparison among the use of ANI, BnA, ANIBr, and BnABr.

	ANI	BnA	ANIBr	BnABr
Can the organic molecule be protonated so that interact with the rest of the species in the MAPbBr ₃ precursor?	No	Yes	Yes	Yes
Does the organic molecule satisfy the geometric condition to form well-defined low-dimensional perovskites?	No	Yes	No	Yes
Does the use of the organic molecule lead to the formation of well-defined low-dimensional perovskites?	No	Yes	No	Yes

In sum, we performed a overall comparison among the use of ANI, BnA, ANIBr, and BnABr and provide its summary to efficiently present the result of our study in Table R1.

Especially, we confirmed the importance of protonation by comparing ANI and ANIBr. On the other hand, the comparison between ANIBr and BnABr revealed the influence of molecular geometry on the formation of perovskites. Based on the results, we can conclude that there is no chance for ANI to compose perovskite lattice, and the geometric factor can be considered only with protonation of the amine when it can interact with the rest of the ionic species. Therefore, we still would like to cling to our claim of which proton transfer is a necessary process to enable BnA to participate in the formation of 3D/2D hybrid perovskites that cannot be achieved without the protonation of BnA. We described the results of the solid-state MAS NMR spectroscopy in the main manuscript.

- R1. Anelli, C. et al. Investigation of dimethylammonium solubility in MAPbBr₃ hybrid perovskite : synthesis, crystal structure, and optical properties. *Inorganic Chemistry* **58**, 944-949 (2019).
- R2. Fergoug, T. & Bouhadda, Y. Determination of Hassi Messaoud asphaltene aromatic structure from ¹H & ¹³C NMR analysis. *Fuel* **115**, 521-526 (2014).
- R3. Li, X. et al. Improved performance and stability of perovskite solar cells by crystal crosslinking with alkylphosphonic acid ω-ammonium chlorides. *Nature Chemistry* **7**, 703–711 (2015).
- R4. Ming Koh, T. et al. Enhancing moisture tolerance in efficient hybrid 3D/2D perovskite photovoltaics. *Journal of Materials Chemistry A* **6**, 2122–2128 (2018).
- R5. Xiao, Z. et al. Efficient perovskite light-emitting diodes featuring nanometre-sized crystallites. *Nature Photonics* **11**, 108–115 (2017).
- R6. Byun, J. et al. Efficient Visible Quasi-2D Perovskite Light-Emitting Diodes. *Advanced Materials* **28**, 7515–7520 (2016).
- R7. Yang, X. et al. Efficient green light-emitting diodes based on quasi-two-dimensional composition and phase engineered perovskite with surface passivation. *Nature Communications* **9**, 570 (2018).
- R8. Yang, X. et al. Effects of Organic Cations on the Structure and Performance of Quasi-Two-Dimensional Perovskite-Based Light-Emitting Diodes. *J. Phys. Chem. Lett.* **10**, 2892–2897 (2019).

Revised parts in the manuscript)

In the page 7, line 6,

To gain an atomic-level insight about the formation of crystalline perovskite, we performed the solid-state magic-angle-spinning (MAS) NMR spectroscopy. Perovskite powders after being scraped off the spin-coated film were used for the measurement. Fig. 2i, j shows the ^1H MAS NMR spectra for the solid-state perovskites with varying BnA concentration (0, 2.4, 30, 50 and 100%) and ANI concentration (0, 2.4, and 100%) under a fast spinning speed of 40 kHz. The pristine sample showed two dominant ^1H peaks at ~ 6.3 ppm and ~ 3.3 ppm which can be assigned to the hydrogens in NH_3 (H-NH $_3$) and CH_3 (H-CH $_3$) of MA^+ , respectively.²⁵ With the addition of BnA, the spectrum had two more peaks that arose from the hydrogens bonded to the benzene ring (H-B) and in CH_2 (H-CH $_2$) of BnA^+ . The positions of H-B varied in the range of ~ 6.7 to ~ 8 ppm and the peak of H-CH $_2$ was at ~ 4.8 ppm.²⁶ The ^1H peak intensity of H-CH $_3$ decreased with an increase in BnA concentration because MA^+ is deprotonated to be methylamine which can be easily evaporated during the film annealing process. Meanwhile, the peak intensities of H-B, H-NH $_3$, and H-CH $_2$ gradually increased confirming that the protonated BnA composes the solid-state crystalline perovskite. Upon the addition of ANI, in contrast, the peak positions of H-NH $_3$ and H-CH $_3$ were invariant indicating ANI did not participate in the formation of perovskite.

In the Fig. 2,

Fig. 2 i, j ^1H MAS NMR spectra (at 14.1 T and a spinning speed of 40 kHz) according to the different amounts of BnA and ANI.

In the Methods.

¹H MAS NMR spectroscopy. ¹H MAS NMR spectra of the samples were collected at room temperature on a Bruker NMR system (14.1 T) at Larmor frequency of 600.41 MHz with a 1.9-mm triple-resonance Bruker NMR probe using a spin-echo pulse sequence at a spinning speed of 40 kHz. Fast sample spinning speed of 40 kHz yielded high-resolution ¹H MAS NMR spectra. The spin-echo pulse sequence ($\pi/2-t-\pi-t$) was used to suppress ¹H background signals from the probe.^{58,59} The $\pi/2$ pulse length of 2 μ s was applied for the samples and a recycle delay time of 10 s was used. Approximately 0.5–6 mg of perovskite powder after being scraped off the film was used. To improve the signal-to-noise ratio, ~16-256 scans were averaged in the ¹H MAS NMR spectra. The ¹H NMR spectra were referenced externally using tetramethylsilane (TMS) solution.

In the Reference.

25. Anelli, C. *et al.* Investigation of Dimethylammonium Solubility in MAPbBr₃ Hybrid Perovskite: Synthesis, Crystal Structure, and Optical Properties. *Inorg. Chem.* **58**, 944–949 (2019).
26. Fergoug, T. & Bouhadda, Y. Determination of Hassi Messaoud asphaltene aromatic structure from ¹H & ¹³C NMR analysis. *Fuel* **115**, 521–526 (2014).
58. Kweon, J. J. *et al.* Evidence from 900 MHz ¹H MAS NMR of Displacive Behavior of the Model Order–Disorder Antiferroelectric NH₄H₂AsO₄. *J. Phys. Chem. C* **119**, 5013–5019 (2015).
59. Kweon, J. J., Fu, R., Steven, E., Lee, C. E. & Dalal, N. S. High Field MAS NMR and Conductivity Study of the Superionic Conductor LiH₂PO₄: Critical Role of Physisorbed Water in Its Protonic Conductivity. *J. Phys. Chem. C* **118**, 13387–13393 (2014).

2) Similarly, the authors continue to refer to the inability of the anilinium system to form a low- dimensional structure due to the difference in basicity. However, the geometry of this system might not be suitable for the formation of a well-defined low-dimensional structure and this must be clearly stated. In fact, an anilinium-based 2D phase could co-exist, yet its poor orientation/crystallinity would prevent it from being

observed experimentally via X-ray diffraction. Considering these factors, the authors should not exclusively ascribe the observed properties to the proton transfer effects.

Answer)

We appreciate the reviewer's valuable comment. We have no doubt that the molecular geometry has to be considered to estimate its eligibility for the formation of perovskites. However, this issue is properly addressed when we make a comparison between ANIBr and BnABr as described in the answer to the reviewer's first comment. The comparison revealed that the anilinium (ANI^+)-mediated phase transformation can occur, but its geometry was not sufficiently suitable for the formation of a well-defined low-dimensional phase. However, neutral ANI which is the main interest in this study would not be the case to account for the molecular geometry because it cannot be protonated in MAPbBr_3 precursor solution, so is not able to be crystallized.

3) The reduced reorientation rate of benzylammonium is ascribed to the π - π interactions and steric repulsion of the neighboring rings. Why does the reorientation impede ion migration? This claim is unclear and requires more clarity.

Answer)

We appreciate the reviewer's valuable comment. We attributed the facilitated ion migration in 3D perovskites to the facile reorientation of methylammonium (MA^+).^{R9} For example, the rotation of MA^+ can lead to the displacement of a halide ion along the direction of the C-N axis of MA^+ based on their attractive Coulomb force. Therefore, the use of less orientationally mobile A-site cation can be expected to impede the halide ion migration. Benzylammonium (BnA^+) in the lattice has the retarded reorientation rate with a low rotational degree of freedom compared to MA^+ owing to their π - π interactions and steric repulsion as a result of neighbouring benzene rings. Thus, the adjacent halide ions to BnA^+ become hard to migrate. We additionally presented a more specific description regarding the issue in the main manuscript in order to clarify the explanation.

R9. Mosconi, E. & De Angelis, F. Mobile Ions in Organohalide Perovskites: Interplay of Electronic Structure and Dynamics. *ACS Energy Letters* **1**, 182–188 (2016).

Revised parts in the manuscript)

In the page 17, line 17,

For example, the rotation of MA⁺ can lead to the displacement of a halide ion along the direction of the C-N axis of MA⁺ based on their attractive Coulomb force. Thus, the incorporation of less orientationally mobile cations with a lower rotational degree of freedom than MA⁺ can reduce ion migration in perovskites.

4) The sentence “3D/2D hybrid dimensional crystal structure formed unlike the conventional quasi-2D perovskites” implies differences between the two structures that have not been defined in the manuscript. This must be appropriately addressed, without inducing further confusion.

Answer)

We appreciate the reviewer’s necessary comment. We decided to remove the last part of the sentence regarding quasi-2D perovskite. Instead, we added a corresponding general description regarding the 3D/2D hybrid dimensional perovskites to avoid potential confusion.

Revised parts in the manuscript)

In the page 4, line 2,

as a result, a 3D/2D hybrid dimensional crystal structure formed in which the formation of 2D perovskite does not degrade the existing 3D phase so that 3D and 2D perovskites coexist.

5) In summary, in accordance with the previous revision, this reviewer is supportive of this study. However, there is no unambiguous evidence that the protonation is the exclusive factor determining the formation of low-dimensional perovskite phases and suppressing the ion migration. Therefore, the “proton-transfer induced 2D/3D hybrid perovskites” term can be misleading, particularly in the title. The authors should thus refrain from using this general language and rather highlight the use of a specific system in this context. For instance, “Benzylamine-Mediated 2D/3D Hybrid Perovskite Formation Reduces Luminance Overshoot” appears more appropriate, without any

need for using the term “extremely” in this context either. Other variants are most welcome as well, this is a very exciting study and this reviewer strives to ensure that it reaches the impact it deserves.

Answer)

We appreciate the reviewer’s constructive comment and genuine effort to help improve the quality of our study. As we believe that our analysis and interpretation regarding the proton transfer are well organized after our revision even with solid-state NMR analysis, we would like to cling to the phrase to “proton-transfer induced 3D/2D hybrid perovskites”, but removed the term “extremely” according to the reviewer’s suggestion.

I believe that it is our responsibility as scientists to contribute to a rigorous scientific practice by using more accurate descriptors instead of coining terms with the objective of publicizing work even at the expense of this being scientifically misleading. This work is already impactful without adding unsubstantiated claims – this reviewer urges the authors to address this carefully to allow the work to express its impact.

Answer)

We appreciate the reviewer’s sincere comment and totally agree with the duty of scientists. We believe that our answer and revised manuscript now satisfy the high standard of *Nature Communications* and the reviewer.

I appreciate your consideration.

REVIEWERS' COMMENTS:

Reviewer #3 (Remarks to the Author):

The authors have carefully addressed this reviewer's remarks and concerns. Even though this might not have been required, it is particularly appreciated that the authors have extended their analysis to atomic-level interactions via solid-state NMR spectroscopy while taking the geometrical aspects into consideration, which provides valuable insights.

It has been a pleasure to review this study and I can recommend it for publication.

I look forward to reading the article in press!

Answers to Reviewers

Reviewer #3

The authors have carefully addressed this reviewer's remarks and concerns. Even though this might not have been required, it is particularly appreciated that the authors have extended their analysis to atomic-level interactions via solid-state NMR spectroscopy while taking the geometrical aspects into consideration, which provides valuable insights.

It has been a pleasure to review this study and I can recommend it for publication.

I look forward to reading the article in press!

Answer)

We appreciate the reviewer's valuable and constructive comments and efforts that significantly helped improve the quality of our study and meet the high standard of *Nature Communications*.